# BAF-1–VRK-1 mediated release of meiotic chromosomes from the nuclear periphery is important for genome integrity

Dimitra Paouneskou [1], Antoine Baudrimont [1], Réka Kelemen [2,4], Marwan Elkrewi [2,4], Angela Graf[1], Shehab Moukbel Ali Aldawla [3], Claudia Kölbl[1], Irene Tiemann-Boege [3], Beatriz Vicoso [2] & Verena Jantsch [1] ✉

Rapid prophase chromosome movements ensure faithful alignment of the parental homologous chromosomes and successful synapsis formation during meiosis. These movements are driven by cytoplasmic forces transmitted to the nuclear periphery, where chromosome ends are attached through transmembrane proteins. During many developmental stages a specific genome architecture with chromatin nuclear periphery contacts mediates specific gene expression. Whether chromatin is removed from the nuclear periphery as a consequence of chromosome motions or by a specific mechanism is not fully understood. Here, we identify a mechanism to remove chromatin from the nuclear periphery through vaccinia related kinase (VRK-1)−dependent phosphorylation of Barrier to Autointegration Factor 1 (BAF-1) in *Caenorhabditis elegans* early prophase of meiosis. Interfering with chromatin removal delays chromosome pairing, impairs synapsis, produces oocytes with abnormal chromosomes and elevated apoptosis. Long read sequencing reveals deletions and duplications in offspring lacking VRK-1 underscoring the importance of the BAF-1−VRK-1 module in preserving genome stability in gametes during rapid chromosome movements.

During gametogenesis the DNA content is halved via two meiotic cell divisions. Crossover (CO) recombination during the first division is crucial for accurate chromosome segregation. Programmed induction of DNA double strand breaks (DSBs) and the preference to repair them via homologous recombination using a chromatid of the other parent as repair template leads to the formation of COs[1,2].

Rapid chromosome movements are a defining feature of meiosis. They facilitate side-by-side-alignment of the parental homologs without entanglement or other undesired chromosome intertwining that could otherwise endanger correct chromosome segregation or cause the formation of undefined chromosome masses. Such movements also mediate installation of the supramolecular structure of the synaptonemal complex (SC) between homologous chromosomes[3–7]. Docking of the chromosome end to the nuclear envelope entails the reconcentration of the SUN protein (SUN-1 in worms) into the chromosome end attachments and aggregation of the KASH protein (ZYG-12 in worms), thereafter movement forces are transmitted through the SUN-KASH bridge from the cytoplasm to the chromosome end attachments. In animal cells, the nuclear lamina underlies the nuclear envelope and confers stability to germline nuclei[8–10]. To prevent the lamina becoming an obstacle to chromosome movements, its physical properties change

[1]Max Perutz Labs, Department of Chromosome Biology, University of Vienna, Vienna BioCenter, Vienna, Austria. [2]Institute of Science and Technology Austria, Klosterneuburg, Austria. [3]Institute for Biophysics, Johannes Kepler University, Linz, Austria. [4]These authors contributed equally: Réka Kelemen, Marwan Elkrewi. ✉e-mail: verena.jantsch@univie.ac.at

to allow for efficient chromosome movements and prevent the formation of aberrant chromosome configurations[9,11].

In interphase cells, the nuclear envelope and associated lamina proteins generate a genome architecture that enables specific gene expression in tissues and at developmental stages[12,13]. Chromatin–nuclear periphery anchorage can be mediated through BAF1 (Barrier to Autointegration Factor 1), a small non-specific DNA-binding protein that connects chromatin to the nuclear periphery through interactions with LEM domain proteins or lamin[14,15]. Tissue-specific BAF-1 interactions with silenced gene loci have been reported in worms, whereas expressed genes were largely devoid of BAF-1[16]. At entry to mitosis, vaccinia-related kinase 1 (VRK1) phosphorylates BAF, which reduces its affinity for chromatin and dissociates chromatin–nuclear envelope interactions[17,18].

Likewise, in *Drosophila* NHK-1 (the VRK1 homolog) removes chromatin from the nuclear envelope to enable karyosome formation in late prophase I[19]. BAF1 dephosphorylation by PP2A promotes nuclear envelope reformation and coherent segregation of the chromosome complement into a single nucleus at the end of mitosis[20–22].

A strong correlation between chromatin reorganization and chromosome movements has been observed in movement-defective genetic backgrounds (e.g[5,23–25].). However, it remains unclear whether additional dedicated mechanisms to remove chromatin from the nuclear periphery cooperate with chromosome movements to mediate side-by-side alignment at that stage of meiosis. No such mechanisms have been reported until now. Under conditional VRK-1 depletion, we find that chromatin remains tethered to the nuclear periphery during chromosome alignment in leptonema–zygonema. This is associated with delayed chromosome pairing, aberrant SC formation, and the formation of oocytes with aberrant chromosomes (including fragments). Genome sequencing of progeny resulting from meiosis in which VRK-1 has been depleted at the chromosome movement stage uncovers newly generated genome variants, including large duplications and deletions. These findings highlight a crucial role for VRK-1 during chromosome movement to ensure genome integrity. This study supports a model where chromosomes have to be removed from the nuclear periphery to facilitate chromosome pairing, synapsis and accurate meiotic recombination and to maintain genomic stability during a meiotic recombination.

## Results

### VRK-1 regulates chromosome organization and tethering in oocytes

*vrk-1*-null mutants have degenerated gonads containing few nuclei that arrest in pachynema[26,27], and so do not complete meiosis or produce offspring. Therefore, we employed an auxin-inducible degradation (AID) system to investigate the effects of VRK-1 depletion at the different meiotic stages of prophase[28], and generated the *vrk-1(syb2608 vrk-1::AID::ha); tir-1::mRuby* (hereafter called *vrk-1::AID::ha*) strain. In this strain, *tir-1* is under the control of the *sun-1* promoter, and with this is expressed in the germline and early embryos. Embryos derived from *vrk-1::AID::ha* oocytes had a similar viability rate to wild type embryos (mean ± SD: 98.1% ± 2.5% and 99.8% ± 0.3%, respectively; Supplementary Table 1), confirming that the HA and the degron tags do not interfere with protein function.

The distal part of the *Caenorhabditis elegans* gonad (i.e. the progenitor zone) serves as a pool of proliferating germ cells. Constant replenishment with germ cells forces the meiocytes to move through the gonad tube at a rate of around one cell row per hour, while passing through the stages of prophase I[29,30] (Fig. 1a, top). We analyzed the effect of VRK-1 depletion on the migration rate of germline nuclei by exposing *vrk-1::AID::ha* worms to ethanol (solvent control) or auxin, followed by incubation in 5-ethynyl-2′-deoxyuridine (EdU) to label nuclei in the progenitor zone. After 18 and 48 hours (h), we compared the percentage of EdU-positive nuclei in all zones in wild type and VRK-

1-depleted gonads and found no difference for zones 1-6 (Supplementary Fig. 1a, b). Zone 7 (late pachynema) had fewer positive nuclei after 48 h, probably as a consequence of VRK-1 depletion in the mitotic compartment. After 18 h, EdU-labeled nuclei had not reached zones 6 and 7 (mid and late pachynema; Fig. 1a, upper and middle germline). Therefore, we used this population of nuclei to assess the effects of VRK-1 depletion from meiotic entry until pachynema on chromosome pairing, synapsis, and CO formation (Fig. 1a, middle germline). A population of nuclei that were exposed to auxin after meiotic entry was seen in diakinesis after 48–50 h (Fig. 1a, lower germline).

Next, we examined VRK-1 localization. A diffuse nuclear, chromatin-associated HA signal was detected in all gonad cells of control *vrk-1::AID::ha* worms (Supplementary Fig. 1c). Immuno-fluorescence against lamin (LMN-1) and quantifications of fluorescence intensities confirmed that VRK-1 overlaps with chromatin and resides within the nuclear envelope in the progenitor zone and transition zone (Supplementary Fig. 1c–e). The strongest HA signal was seen in progenitor zone nuclei where, in the few dividing nuclei, it became associated with the nuclear rim marked by LMN-1 (Supplementary Fig. 1c, white arrows). VRK-1 depletion led to the effective loss of signal. After 6 h of auxin treatment, HA signal intensity was comparable in *vrk-1::AID::ha* and untagged wild type nuclei (N2) (Supplementary Fig. 1f–h). These data confirm that VRK-1 is a chromatin associated protein and the AID system is an efficient tool to generate conditional VRK-1 knockouts.

After meiotic entry, SUN-1 (nuclear envelope protein) becomes phosphorylated[31,32], and chromatin is reorganized and appears half-moon shaped[33]; both coincide with rapid chromosome movements in prophase. In *vrk-1::AID::ha* auxin treated germlines, we observed a striking reduction in the proportion of nuclei with clustered chromatin after meiotic entry (Fig. 1b, dashed circled nuclei indicate nuclei with clustered chromatin vs arrowhead marked nuclei which indicate nuclei with non-clustered chromatin). Following VRK-1 depletion for 6 h, we quantified the percentage of phospho-SUN-1-positive nuclei with clustered chromatin in the first 10 cell rows after meiotic entry. Auxin treatment led to a two-fold reduction in half-moon-shaped nuclei (29.4% ± 8.1%; compared with control, 48.5% ± 6.4%) after 6 h (Fig. 1c and Supplementary Fig. 1f, middle panel), suggesting a defect in chromatin reorganization upon meiotic entry. VRK-1 depletion did not prolong the SUN-1 (S8Pi) positive zone (Supplementary Fig. 1i), indicating progression through the gonad and the downstream meiotic events were not affected by the 6 h depletion. To examine whether pairing center (PC) protein recruitment to the chromosome ends after VRK-1 depletion was affected, we conducted immunofluorescence against SUN-1 (S8Pi) and PLK-2. In auxin treated *vrk-1::AID::ha* transition zone meiocytes, SUN-1 patches (at nuclear envelope tethered chromosome ends) colocalized with PC-recruited PLK-2 similarly to the ethanol treated germlines (Fig. 1d).

Nuclear lamina mutants that stiffen the lamina have defective chromatin reorganization and decelerated chromosome end-led movements[23]. Therefore, we monitored the mobility of chromosome ends in *vrk-1::AID::ha* germline nuclei by tracking the movement of SUN-1::GFP, which becomes concentrated in chromosome ends that are associated with the nuclear envelope. However, no significant difference in the speed of chromosome displacement tracks, distance traveled, or area covered were found after VRK-1 depletion (Supplementary Fig. 2a–c). In summary, VRK-1 depletion interferes with chromosome reorganization during the leptotene–zygotene stage but does not affect the mobility and localization of chromosome ends to the PCs.

To further explore how chromatin architecture is altered in *vrk-1::AID::ha*, we performed multiplexed DNA fluorescence in situ hybridization (FISH) to simultaneously visualize chromosomes I, II, and III (linkage groups LGI, LGII, LGIII)[34] (Fig. 1e, f). High resolution imaging of the three chromosomes in the transition zone suggested a more

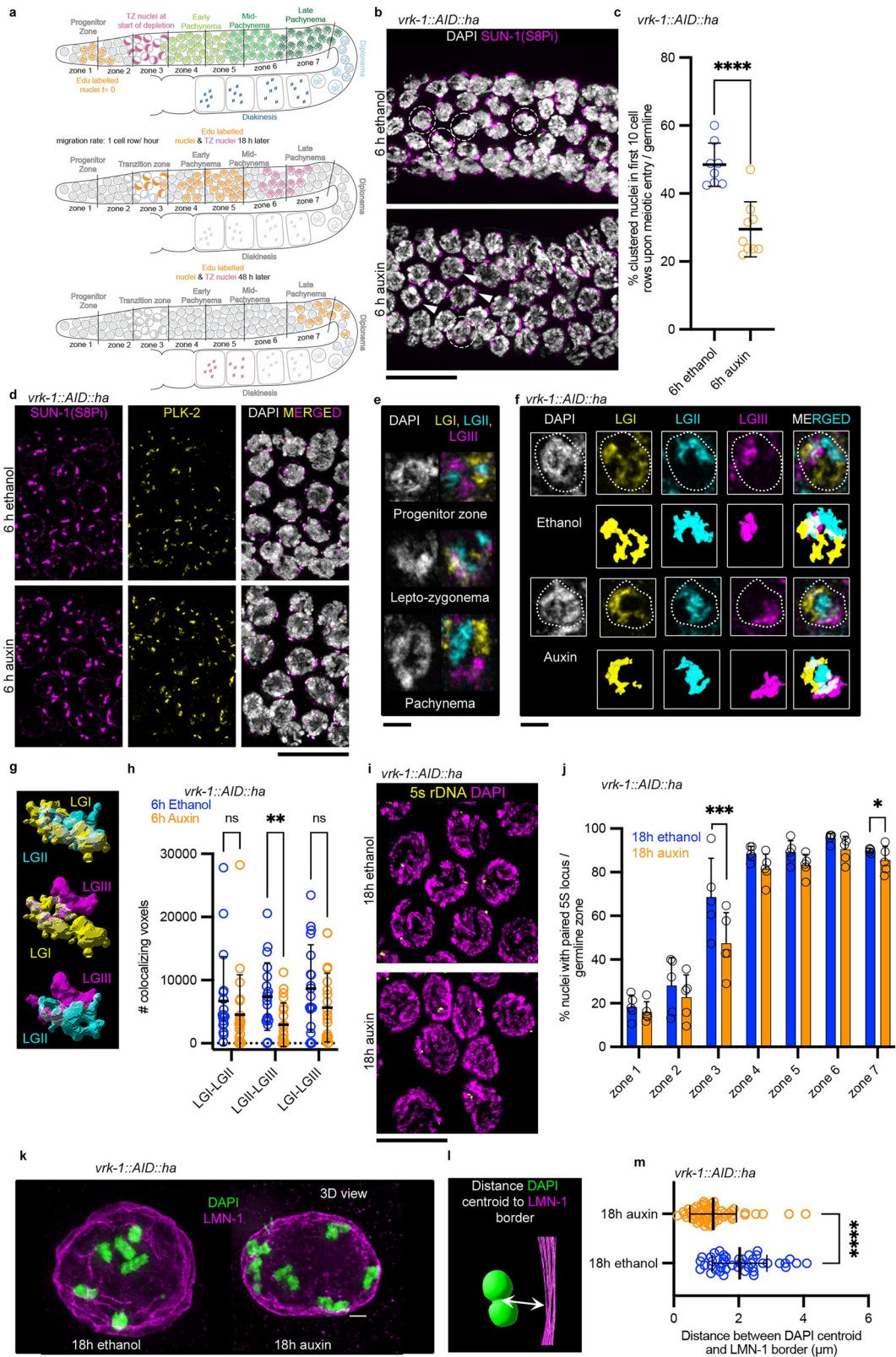

pronounced association to the nuclear periphery after auxin treatment (Fig. 1f). We measured the three-dimensional (3D) overlap between the identified territories (Fig. 1g, h; see Supplementary methods). VRK-1 depletion significantly reduced the overlap between chromosomes LGII and LGIII (adjusted $P = 0.0101$, Multiple Mann-Whitney tests) associated with large effect size (Cliff's delta = -0.531), with a smaller effect for the overlaps between LGI and LGII and between LGI and LGIII

(Cliff's delta = -0.321, adjusted $P = 0.1614$, and Cliff's delta = -0.266, adjusted $P = 0.1614$, Multiple Mann-Whitney tests, respectively) indicating variations in colocalization patterns (Fig. 1g, h). Additional FISH experiments using the 5S ribosomal probe consistently showed delayed chromosome pairing in auxin treated *vrk-1::AID::ha* nuclei after meiotic entry (Fig. 1i, j). Thus, the multiplexed FISH and pairing FISH data demonstrate that VRK-1 depletion correlates with chromatin

**Fig. 1 | VRK-1 depletion impedes chromosome reorganization and tethering in early meiosis. a** Schematic of the *C. elegans* germline divided into seven equal zones for quantification purposes. Progenitor zone nuclei undergoing replication are labeled with EdU (orange, t = 0). Middle and bottom: Migration rate of of EdU-positive nuclei (orange) and transition zone nuclei (magenta) 18 h and 48 h after labeling respectively. EdU = 5-ethynyl-2′-deoxyuridine. **b** Insets from *vrk-1::AID::ha* germlines stained with DAPI (white) and anti-PiSer8 SUN-1 antibody (meiotic entry marker; magenta) after 6 h in ethanol or auxin. Dashed circles indicate clustered chromatin nuclei and arrowheads non-clustered chromatin nuclei. Scale bar: 10 μm. HA = hemmaglutinin **c** Percentage of nuclei with clustered chromatin in the first 10 meiotic cell rows in *vrk-1::AID::ha* germlines. Mean ± SD: 6 h ethanol, 48.5% ± 6.4%, *n* = 8 germlines; 6 h auxin, 29.5% ± 8.1%, *n* = 9 germlines; *P* < 0.0001 Two-sided Fischer's exact test. **d** Transition zone insets of *vrk-1::AID::ha* germlines stained with PiSer8 SUN-1 (magenta), PLK-2 (yellow) and DAPI (merged). Scale bar: 10 μm. **e** Insets of nuclei from germlines stained with DAPI (white) and analyzed by multiplexed FISH for linkage group (LG) I (yellow), II (cyan), and III (magenta) for the indicated meiotic stages. Scale bar 2: μm. **f** As in (**e**) for the indicated conditions. A

3D view of each LG is also depicted. Scale bar: 2 μm. **g** Representative 3D analysis of overlapping chromosomes. **h** Quantification of the degree of overlap between different LGs for the indicated conditions. Mean ± SD, n number and exact *P*-values (calculated with Multiple Mann–Whitney two-sided tests applying the Holm-Šídák method, and Cliff's delta for size effect) are included in the Source table. **i** Insets from *vrk-1::AID::ha* germlines analyzed by FISH (against 5S rDNA; yellow) and stained for DAPI (magenta). Scale bar: 10 μm. **j** Percentage of paired nuclei per germline zone for the indicated genotype and conditions (*n* = 5 germlines/condition). FISH = Fluorescence in situ hybridization. %Mean ± SD and exact *P*- values (two-sided Fischer's exact test) per zone in Supplementary Table 2. **k** 3D view of *vrk-1::AID::ha* diakinesis oocytes for the indicated conditions. Scale bar: 2 μm. **l** Graphic representation of the distance measured in 3D between the center of a DAPI body (centroid, green) and the oocyte border (purple). **m** Quantification of the DAPI centroid–oocyte border distance (marked by LMN-1) for the indicated conditions. Mean ± SD, n number and *P*-value calculated with Mann–Whitney test two-sided in Source table.

tethering to the nuclear periphery, a lower degree of chromosome overlap, and, subsequently, a reduction in chromosome configurations indicative of an ongoing alignment process.

Next, we depleted VRK-1 throughout the meiotic time course (18 h) and analyzed chromosomes in diakinesis nuclei. In the wild type, diakinesis nuclei have six bivalents, corresponding to connected condensed parental chromosome pairs that are ready for metaphase I[33]. We investigated the position of bivalents at diakinesis in the -1 oocyte by co-staining with DAPI (chromosomes) and anti-LMN-1 antibody (nuclear periphery; Fig. 1k, l) and measured the distance from each DAPI body (centroid) to the nuclear envelope. In the wild type, the six bivalents were a mean distance of 2 μm from the nuclear envelope (Fig. 1m). In contrast, under VRK-1 depletion bivalents were a mean distance of 1.2 μm from the nuclear periphery (Fig. 1m), indicating chromosome anchorage. Similarly, in *Drosophila* depletion of NHK-1 (the ortholog of VRK-1) inhibits karyosome formation and chromosomes remain attached to the nuclear periphery[19].

In summary, our data demonstrate that VRK-1 plays an essential role in chromosome reorganization by promoting the release of the chromatin from the nuclear periphery, especially during the leptotene–zygotene and diakinesis stages of meiosis, in the *C. elegans* germline.

## BAF-1 is the relevant target for VRK-1 kinase to mediate chromosome reorganization

In mitosis, VRK-1-mediated phosphorylation of BAF-1 leads to the dissociation of chromatin from the nuclear periphery[17,18,20,21,35,36]. To investigate whether BAF-1 is a VRK-1 target during meiotic chromosome reorganization, we depleted *baf-1* by RNAi in auxin-treated or control *vrk-1::AID::ha* worms and quantified the percentage of nuclei with clustered chromatin in the first 10 cell rows after meiotic entry (SUN-1 Ser8Pi antibody labeling; Fig. 2a). Compared with the wild type, VRK-1 depletion reduced the number of nuclei with clustered chromatin (Fig. 2a, b), whereas *baf-1* depletion (*baf-1 RNAi*/ethanol) did not (Fig. 2a, b). However, when both VRK-1 and *baf-1* were depleted, the percentage of nuclei with clustered chromatin was similar to in the wild type (Fig. 2a, b). Depletion of both *baf-1* and VRK-1 also released tethered chromosomes from the nuclear periphery in the -1 diakinesis oocytes, rescuing the peripheral anchorage induced upon VRK-1 depletion (Fig. 2c, Supplementary Fig. 2d). Therefore, *baf-1* depletion in *vrk-1::AID::ha* auxin treated cells restores chromosome reorganization in the transition zone and untethers diakinesis chromosomes from the nuclear periphery.

We then examined BAF-1 localization before and after VRK-1 depletion using a *flag::baf-1* worm strain (for functionality, see Supplementary Table 1). Immunostained germlines showed BAF-1 nuclear enriched signal throughout the germline (Supplementary Fig. 2e,

+VRK-1). Further quantification of signal intensities confirmed the rather weak BAF-1 localization to chromatin in mitotic and transition zone nuclei in the wild type (Fig. 2d–e, ethanol). However, VRK-1 depletion globally increased the BAF-1 signal both at the nuclear periphery and on chromatin (Supplementary Fig. 2e, -VRK-1), with a two-fold increase in transition zone nuclei (Fig. 2d–e, auxin).

In vivo and in vitro studies have shown that the release of BAF-1 from chromatin is mediated by VRK-1 phosphorylation at highly conserved residues: Thr3 and Ser4[14–17] (Supplementary Fig. 2f). Therefore, we generated two phospho-mutants (*flag::baf-1^{T3A}* and *flag::baf-1^{S4A}*) that exhibited distinct phenotypes (Supplementary Table 1): *flag::baf-1^{T3A}* was viable with a normal brood size (mean ± SD: 99% ± 1.1 and 231 ± 59.2 respectively), whereas *flag::baf-1^{S4A}* was embryonic lethal, with a very low brood size (mean ± SD: 31.8 ± 34.6). We found no significant difference in progenitor zone length (normalized to gonad length) for *flag::baf-1^{T3A}* or *flag::baf-1^{S4A}* compared with wild type (Supplementary Fig. 2g). The percentage of nuclei with clustered chromatin in the first 10 cell rows after meiotic entry was similar for both *flag::baf-1* and *flag::baf-1^{T3A}*, but significantly lower in *flag::baf-1^{S4A}* (Fig. 2f, g). Live imaging of SUN-1::GFP aggregates showed no significant difference in the speed of chromosome displacement in *flag::baf-1^{S4A}* and *flag::baf-1* germlines (Supplementary Fig. 2h).

We next generated an anti-BAF-1 Ser4Pi antibody. The antibody's specificity was confirmed since Western blot analysis of whole animal extracts from *flag::baf-1* and *flag::baf-1^{T3A}* worms revealed the BAF-1 Ser4 Pi band, which was absent in the *flag::baf-1^{S4A}* worm extracts (Supplementary Fig. 2i). Further Western blot analysis of biochemical fractions of *vrk-1::AID::ha; flag::baf-1* germline nuclei showed that a BAF-1 Ser4 Pi band was absent in auxin *vrk-1::AID::ha*, unlike in wild type nuclei (Fig. 2h white and black arrows). This confirmed that BAF-1 Ser4 is a VRK-1 target in *C. elegans* germline nuclei.

These combined data provide evidence that BAF-1 Ser4 phosphorylation is crucial for germline development and meiosis. Mutation of BAF-1 Ser4 mimicked VRK-1 depletion regarding chromosome reorganization but did not affect chromosome end-led mobility.

## VRK-1 depletion leads to meiotic defects

Given the aberrant chromosome reorganization and the delayed chromosome pairing in auxin treated *vrk-1::AID::ha*, we next examined whether synapsis and CO formation are normal in VRK-1-depleted germline nuclei. After 18 h of auxin treatment, transition zone (but not progenitor zone) nuclei had reached gonad zones 6 and 7 (Fig. 3a). We assessed synapsis formation by labeling chromosome axes and the central element of the SC with antibodies against HTP-3 and SYP-1, respectively[37,38]. In the wild type, SYP-1 expression directly follows HTP-3 expression, with the HTP-3 and SYP-1 signals overlapping in

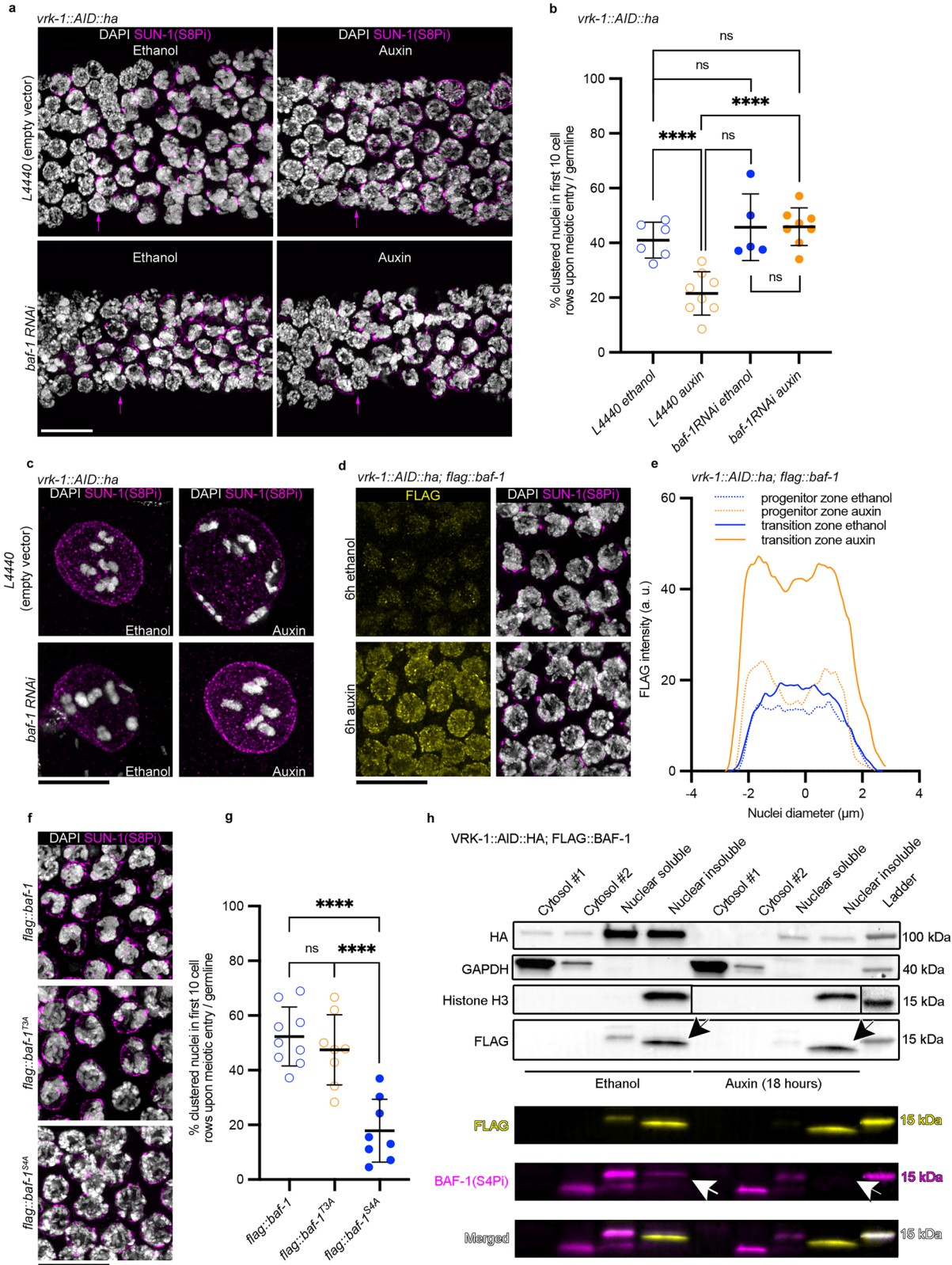

pachynema (Fig. 3b and Supplementary Fig. 3a). Quantification of synapsis kinetics (Fig. 3c and Supplementary Fig. 3a) revealed that in auxin treated *vrk-1::AID::ha* synapsis assembly was significantly delayed (Fig. 3c, zones 3–6 corresponding to transition zone until mid-pachynema) and, unlike in the wild type, was not complete by late pachynema (Fig. 3b, c, zone 7). Consistent with this, synapsis was significantly impaired in *flag::baf-1^{S4A}* compared with *flag::baf-1*,

showing that inability to phosphorylate BAF-1 Ser4 results in severe synapsis failure (Supplementary Fig. 3b).

Prophase movements were previously shown to be sustained in synapsis mutants, with PLK-2 remaining at chromosome end attachments[39,40]. In the wild type, PLK-2 localized to aggregates at the nuclear envelope in transition zone nuclei and relocalized to chromosome axes in early pachynema (Supplementary Fig. 3c, upper

**Fig. 2 | Impaired chromosome reorganization upon VRK-1 depletion is mediated by BAF.** **a** Insets from *vrk-1::AID::ha* germlines stained with DAPI (white) and anti-PiSer8 SUN-1 antibody (magenta) after depletion of VRK-1, *baf-1*RNAi, or both. Magenta arrows delineate meiotic entry. Scale bar: 10 μm. **b** Percentage of clustered chromatin nuclei in the first 10 meiotic cell rows in the *vrk-1::AID::ha* germlines. Mean ± SD, L4440 (empty vector)–ethanol: 41.0% ± 6.5%; *n* = 6 vs L4440–auxin: 21.5% ± 7.9%; *n* = 8, *P* < 0.0001 Two-sided Fisher's exact test; *baf-1* RNAi–ethanol: 45.7% ± 12.1%; *n* = 5, *P* = 0.3663, Fisher's exact test compared with L4440–ethanol; *baf-1* RNAi–auxin: 45.9% ± 6.9%; *n* = 8, *P* = 0.3103, Fisher's exact test compared with L4440–ethanol. n= number of germlines. **c** DAPI staining (white) and PiSer8 SUN-1 immunostaining (magenta) in -1 oocytes of *vrk-1::AID::ha* germlines. Scale bar: 10 μm. **d** Transition zone insets from *vrk-1::AID::ha; flag::baf-1* germlines treated with ethanol (top) or auxin (bottom) and immunostained for FLAG (yellow) and co-stained with DAPI (white). Scale bar: 10 μm. **e** Average line profile analysis of FLAG signal intensity in the center of the nucleus for the indicated treatments: ethanol, *n* = 63 nuclei for progenitor zone, 71 for transition zone; and auxin, *n* = 94 for progenitor zone, 93 for transition zone. **f** Transition zone insets from *flag::baf-1*, *flag::baf-1^T3A^*, *flag::baf-1^S4A^* germlines immunostained for PiSer8 SUN-1 (magenta) and co-stained with DAPI (white). Scale bar: 10 μm. **g** Percentage of clustered chromatin nuclei in the first 10 meiotic cell rows of germlines in the indicated genotypes. Mean ± SD, *flag::baf-1*: 52.3% ± 10.8%, *n* = 9 germlines; *flag::baf-1^T3A^*: 47.4% ± 12.8%, *n* = 8 germlines; *P* = 0.1217; *flag::baf-1^S4A^*: 17.9% ± 11.5%, *n* = 8 germlines; *P* < 0.0001 two-sided Fisher's exact test). **h** Western blot of the indicated cellular fractions from *vrk-1::AID::ha* worm pellets for the indicated genotypes treated with ethanol or auxin. Arrows indicate the presence (in ethanol) or absence (in auxin) of the BAF-1 S4Pi band. Full blots included in Source table. Anti-GAPDH and anti-H3 antibodies were used as loading controls. GAPDH = glyceraldehyde-3-phosphate dehydrogenase, H3 = Histone 3, kDA = KiloDalton.

germline). Consistent with the synapsis delay in auxin treated *vrk-1::AID::ha*, PLK-2 aggregation on chromosome ends was significantly prolonged and relocalization to the axes was delayed (Supplementary Fig. 3c, d).

PLK-2 relocalization to the SC is postulated to occur in response to CO designation[41]. Therefore, we investigated CO efficiency in VRK-1-depleted and control germlines using MSH-5 as a readout for developing COs (Fig. 3d). In response to VRK-1 depletion, MSH-5 recruitment was significantly delayed in late transition zone–early pachynema (Fig. 3e, 21–30 cell rows; for quantification details, see Supplementary methods), resulting in a significantly higher number of MSH-5 foci in mid and late pachynema (cell rows 11–20 and 1–10, respectively; Fig. 3d, e). Retraction of the CO factor ZHP-3[42,43] from the SC to CO foci is an indicator of CO maturation. ZHP-3 retraction was significantly delayed in *vrk-1::AID* germlines (Supplementary Fig. 3e, f).

Incomplete synapsis can trigger the elimination of affected oocytes by apoptosis[44] in late pachynema–early diplonema[45]. This prompted us to examine whether apoptosis was increased in nuclei devoid of VRK-1 in transition zone (Fig. 3f, top). SYTO12 staining showed an increased number of apoptotic corpses after VRK-1 depletion compared with the wild type (Fig. 3f).

In summary, in absence of VRK-1, SC formation is slower (and fails to complete in a few chromosomal regions), CO formation is delayed, and programmed cell death in the germline is elevated.

### VRK-1 depletion causes aberrant chromosomes in diakinesis oocytes

To determine the effect of chromatin "overtethering" in transition zone nuclei on the formation of connected bivalents, we depleted VRK-1 for 50 h to allow transition zone nuclei to reach diakinesis (Fig. 4a). Wild type (ethanol treated) oocytes had six DAPI-stained bodies (mean ± SD, 6 ± 0.2), corresponding to the six pairs of homologous chromosomes (Fig. 4b, upper left panel, and Fig. 4c). Auxin treated *vrk-1::AID::ha* oocytes had significantly more DAPI bodies (mean ± SD, 7.7 ± 1.57; Fig. 4b, lower left panel, and Fig. 4c). In contrast, this was not seen in nuclei that had been depleted of VRK-1 in late pachynema (see below). The increased number of DAPI bodies was dependent on meiotic DSB induction and CO formation between homologs. Auxin treated *vrk-1::AID::ha; spo-11 (ok79)* and *vrk-1::AID::ha; msh-5 (me23)* oocytes contained 12 DAPI bodies on average, similar to ethanol-treated *spo-11 (ok79)* (Fig. 4b, middle panel, Fig. 4c) and *msh-5 (me23)* in ethanol (Fig. 4b, right panel, Fig. 4c). Blocking alternative repair pathways such as non-homologous end joining (NHEJ; i.e. in double mutants *vrk-1::AID; cku-70(tm1524)* or *vrk-1::AID; lig-4(ok716)*[46]) slightly reduced the number of DAPI bodies (Supplementary Fig. 4a). Combining treated *vrk-1::AID::ha* with the *mus-81* or *polq-1* mutant to inhibit the MUS-81 nuclease, which can resolve joint DNA structures[47–49], or of the micro homology mediated end joining pathway[50,51] significantly reduced the number of DAPI bodies (Supplementary Fig. 4b). We also noticed the presence of intrachromosomal bridges after VRK-1

depletion (approximately 2.3 bridges per oocyte; Fig. 4b, arrow in lower left inset, and Fig. 4d), but hardly any in the wild type (Fig. 4d). The number of bridges was slightly decreased in *vrk-1::AID::ha; lig-4(ok716)* oocytes, but not in double mutants with components of the other repair pathways (Supplementary Fig. 4c, d). In contrast, knocking out *spo-11* or *msh-5* in *vrk-1::AID::ha* auxin treated oocytes fully suppressed the formation of intrachromosomal bridges (Fig. 4d), indicating that the formation of bridges fully depends on the repair of meiotic DSBs. We conclude that once DSBs and COs are being made, VRK-1 becomes essential for chromosomal integrity in diakinesis, and alternative repair pathways (*mus-81* and *polq-1* dependent) operate upon VRK-1 depletion in an unscheduled manner.

To better characterize the aberrant bivalent structures, we used the *dtn-1::GFP* transgene to visualize telomeric sequences[52]. Under ethanol conditions, each condensed bivalent displayed eight GFP signals, corresponding to the telomeres of the individual chromatids (Fig. 4e–g, schematic and ethanol conditions). In contrast, a significantly higher proportion of auxin treated *vrk-1::AID::ha* oocytes contained DAPI bodies with fewer than eight GFP signals, indicating DNA fragmentation (Fig. 4f, circle in auxin conditions and Fig. 4g). Intrachromosomal bridges, depicted by bivalents with eight GFP signals connected by a DNA bridge (Fig. 4f, arrow in auxin conditions and Fig. 4g) were observed in almost all VRK-1 depleted oocytes, but in only a quarter of wild type oocytes (Fig. 4g). Interchromosomal bridges were rarely detected in either the presence or absence of VRK-1 (0.03% in *vrk-1::AID::ha* auxin treated oocytes) (Fig. 4g).

Surprisingly, chromosome bridges appeared with the same penetrance when oocytes were depleted of VRK-1 in diplonema (when most of the recombination process is completed) or late pachynema. In diakinesis at 6 and 24 h after VRK-1 depletion (Supplementary Fig. 4e, f), the number of DAPI bodies was similar to the wild type (Supplementary Fig. 4g), but the number of intrachromosomal bridges was as high as the number observed at the 50-hour timepoint (Supplementary Fig. 4h). Formation of these bridges was also dependent on SPO-11 and MSH-5 (Supplementary Fig. 4i). Depletion of *baf-1* by RNAi in auxin treated *vrk-1::AID::ha* oocytes (for details see Methods) significantly reduced the number of intrachromosomal bridges to almost wild type levels (Fig. 4h, i).

In summary, these data show that VRK-1/BAF-1-mediated chromatin overtethering compromises chromosome integrity in the wake of meiotic recombination, leading to chromosome fragmentation, and that alternative repair pathways are switched on. BAF-1-dependent intrachromosomal bridges are formed in diplonema.

### Transient VRK-1 loss in early meiosis yields heritable genomic variants

We found that VRK-1 depletion causes ectopic chromatin–nuclear periphery tethering during prophase. Therefore, to understand the consequences of temporary "overtethering" of chromosomes during pairwise alignment, we depleted VRK-1 for 18 h and then transferred

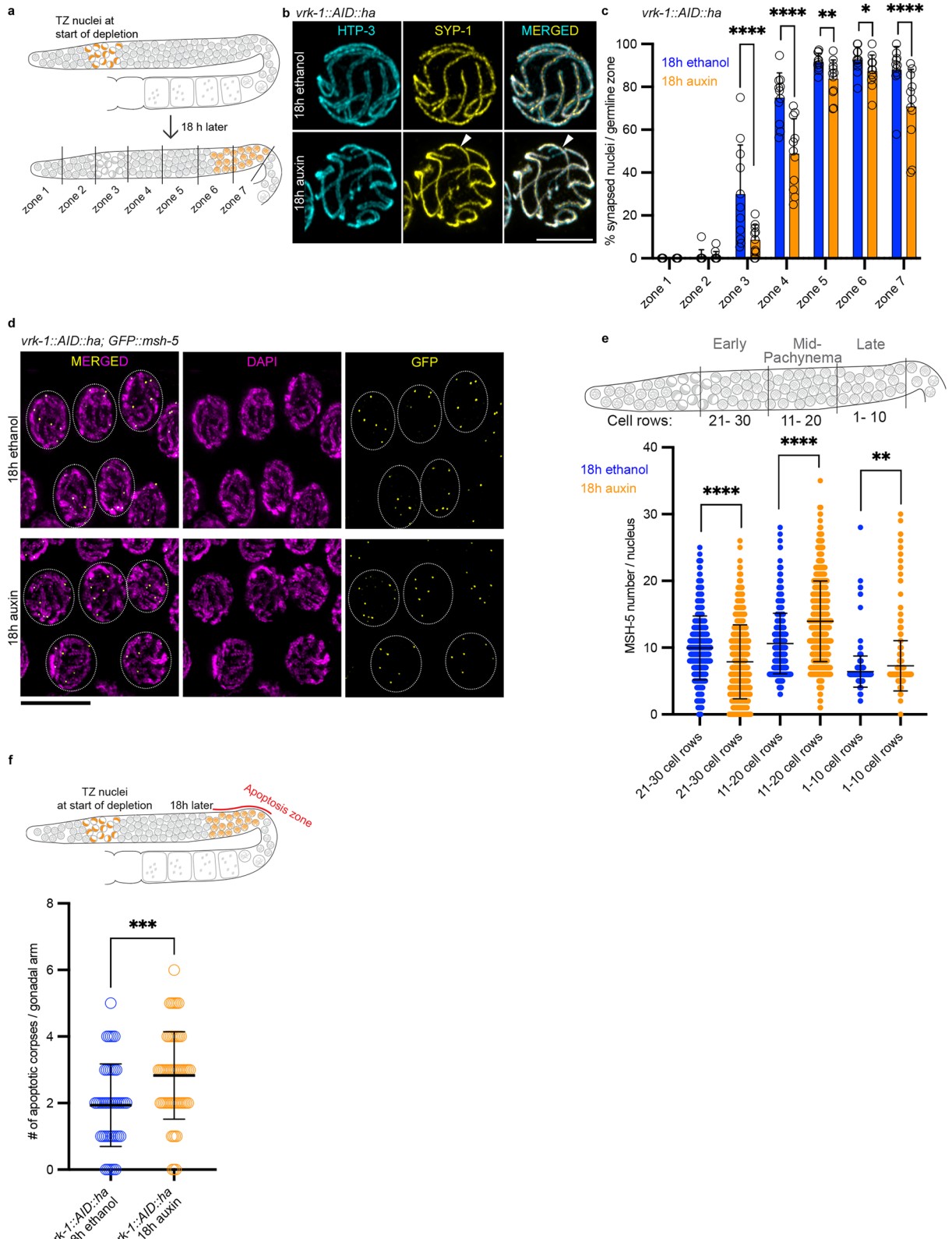

the worms to regular plates for approximately 45 h. This strategy enabled VRK-1 to be expressed from pachynema onwards and throughout the embryonic divisions (Fig. 5a). Within this experimental system, transition zone–early pachytene meiocytes that had undergone VRK-1 depletion were able to develop into fertilized embryos (Fig. 5a). By collecting only those embryos formed in the last 8 h (interval 56 – 63 h, Fig. 5a), we ensured that (1) they derived from

oocytes that had been depleted of VRK-1 in the transition zone–early pachynema stage (for experimental details see Supplementary Methods) and (2) VRK-1 expression after this stage allowed for normal embryo development. We confirmed that by the time of sample collection (day 3) embryo viability had vastly recovered to wild-type levels (Supplementary Fig. 5a) whereas no increase of offspring males was detected (Supplementary Fig. 5b). VRK-1 levels in late pachynema were

**Fig. 3 | Chromatin "overtethering" upon VRK-1 depletion impedes important meiotic processes. a** Schematic representation of germlines showing the migration rate of transition zone nuclei at 18 h after VRK-1 depletion. **b** High-resolution images of *vrk-1::AID::ha* nuclei stained for HTP-3 (cyan) and SYP-1 (yellow) after 18 h in ethanol or auxin. Arrowheads indicate the unsynapsed region stained only with HTP-3. Scale bar: 5 μm. **c** Percentage of fully synapsed nuclei per germline zone for the indicated conditions and genotype. Mean ± SD per zone, 18 h ethanol vs 18 h auxin: zone 1: 0.0% ± 0.0% for both; zone 2: 0.9% ± 3.0% vs 0.9% ± 2.2%, $P = 0.7669$; zone 3: 29.9% ± 22.9% vs 8.5% ± 7.0%, $P < 0.0001$; zone 4: 74.9% ± 11.6% vs 48.6% ± 16.4%, $P < 0.0001$; zone 5: 91.6% ± 4.0% vs 83.7% ± 8.8%, $P = 0.0026$; zone 6: 92.5% ± 6.0% vs 87.2% ± 7.6%, $P = 0.0368$; and zone 7: 87.8% ± 10.9% vs 70.7% ± 17.4%, $P < 0.0001$. Two-sided Fisher's exact test was used to assess statistical significance, with 11 germlines analyzed for each condition. **d** Late pachynema insets from *vrk-1::AID::ha; GFP::msh-5* germlines, showing GFP signals (yellow) and DAPI staining (magenta) after exposure to the indicated conditions. Scale bar: 10 μm. **e** Top: schematic representation of a germline, with the indicated cell rows. Bottom: number of GFP::MSH-5 foci per nucleus for the indicated treatments (ethanol, $n = 9$ germlines; auxin, $n = 11$ germlines) and germline rows. Mean ± SD per zone (proximal to diplonema), 18 h ethanol vs 18 h auxin: zones 21–30, 10 ± 4.8 vs 7.9 ± 5.5, $P < 0.0001$; zones 11–20, 10.6 ± 4.5 vs 13.9 ± 6, $P < 0.0001$; zones 1–10, 6.4 ± 2.3 vs 7.3 ± 3.8, $P = 0.0031$. The Two-sided Mann–Whitney test was used to assess statistical significance. **f** Top: schematic representation of a germline, showing that 18 h after VRK-1 depletion, transition zone nuclei reach the region of physiological apoptosis. TZ = transition zone. Bottom: quantification of apoptotic corpses for the indicated genotype and conditions. Mean ± SD: 18 h, ethanol (n = 47): 1.9 ± 1.2 vs auxin (n = 59): 2.8 ± 1.3; $P = 0.0004$, Two-sided Mann–Whitney test. Each datapoint represents a germline.

also similar to the wild type following VRK-1 resynthesis (Supplementary Fig. 5c). DNA was extracted from auxin-treated and control embryos and sequenced using Oxford Nanopore Technology (ONT). As the two libraries had different total read numbers and length distributions, we resampled the reads obtained from both samples to eliminate bias prior to running the comparison (Fig. 5b; see Supplementary methods for details). We then compared the number of structural variants (SVs), including deletions, duplications, insertions, inversions and translocations. Since genome variants arise independently in each oocyte germline nucleus, we reasoned that only a few of these events would be captured with sequencing coverage of 30 times. Therefore, we focused on SVs supported by only one or two reads. Because ONT sequencing has a very high rate of errors consisting of short insertions and deletions[53], we focused on larger SVs, larger than 1000 bp.

VRK-1 depletion caused a striking overall 2- and 3-fold increase in the number of deletions and duplications respectively which became more prominent when sorted into three classes: 1000–10,000 bp, 10,000–50,000 bp, and 50,000–100,000 bp (Fig. 5c–e, and Supplementary Fig. 6a, b). We also counted the number of SVs in non-overlapping windows of 300,000 bp across the genome and found that both deletions and duplications were significantly enriched throughout the genome after VRK-1 depletion (Supplementary Fig. 6c, d). In contrast, the overall fold change in insertions, inversions, and translocations was only slightly increased, irrespective of size (Fig. 5c and Supplementary Fig. 7a, b), although the window-based comparison showed that some were significantly concentrated to particular regions in the genome (Supplementary Fig. 7c, d); however, those represent only minor fractions of the total reads. Since deletions and duplications showed the largest fold change between samples, we focused on these. The number of large deletions and duplications (>10,000 bp) increased by >6-fold after VRK-1 depletion (Fig. 5d, e), with no specific sequence repeats detected at DNA regions flanking the rearrangements sites (Supplementary Fig. 8). We next assessed whether these variants were enriched in specific chromosomal regions. Deletions and duplications were distributed homogeneously along the chromosomes, with no evidence of enrichment. Nevertheless, their numbers were consistently increased after VRK-1 depletion compared with the wild type (Fig. 5f, g and Supplementary Figs. 9–14). Similarly, insertions and inversions seem to be evenly distributed along the genome (Supplementary Figs. 15–20).

These findings suggest that transient VRK-1 depletion disrupts genome integrity during early meiotic prophase, leading to heritable genomic alterations in the offspring.

## Discussion

This study investigated whether chromosome reorganization in early prophase relies solely on chromosome movements or whether other dedicated mechanisms remove chromatin–nuclear periphery contacts. We suggest VRK-1-mediated phosphorylation of BAF-1 as a possible mechanism for scheduled removal of these contacts in animal cells. BAF-1 was enriched at the nuclear periphery after VRK-1 depletion and chromosome reorganization phenotypes resembled those seen in *baf-1* phospho-mutants. We analyzed the consequences of chromatin "overtethering" to the nuclear periphery on meiotic processes and genome integrity in the offspring (phenotypes shown in Fig. 6). Delayed chromosome pairing, SC formation, and gaps in the SC were identified. The SC gaps may trigger elevated germline cell death by either activating the mechano-sensitive checkpoint[44] or accumulating aberrant recombination intermediates, which trigger the DNA damage checkpoint[54]. The presence of SC-free regions may keep the DSB machinery active for a much longer period and, thus, overwhelm the DNA repair system[55–59]. Alternatively, SC-free regions may reflect interruption to chromosome alignment caused by topologically aberrant chromosomes structures[60]; these might also enhance apoptosis.

In diakinesis chromosomes, we detected an increased number of DAPI bodies, indicating the presence of univalents and DNA fragments. Since their formation depends on SPO-11 and MSH-5, we conclude that aberrant DAPI bodies are derived from meiotic DSBs and not from extra chromosomes being imported into meiosis. Chromosome fragments and univalents may also arise through persistent DSBs and/or persistent recombination intermediates, due to imprecise chromosome pair alignment and SC formation hindering scheduled repair that uses a chromatid from the other parental chromosome as the repair template. Hyper-resection of some persistent DNA ends when the homologous template is absent may trigger the engagement of alternative DNA repair pathways. Elimination of some DNA repair processes that are not usually employed during meiosis (including NHEJ and alternative end joining) altered the DAPI body count, suggesting that VRK-1 depletion leads to the selection of DNA repair pathways that are not normally used in meiosis. The fact that the aberrant DAPI body counts were more pronounced when VRK-1 was depleted at the stage when chromosome movements are most pronounced (50-hour timepoint) than at late pachynema or diplonema (24- and 6-hour timepoints, respectively) suggests that impaired chromosome alignment and/or faulty SC formation is likely responsible for the aberrant DNA repair. Vigorous chromosome movements may hinder the repair of some types of DNA breaks when chromosome alignment is slowed down or imperfect or mechanical forces may tear chromosomes apart when chromosome end-led movements exert too much tension on chromatin that is ectopically attached to the nuclear periphery. Overall, our data show that meiotic recombination can occur in the absence of VRK-1, albeit delayed and associated with elevated numbers of MSH-5-decorated intermediates. The latter might be a consequence of delayed SC formation combined with delayed repair or caused by elevated DSB induction or imperfect "CO counting mechanisms" due to SC interruptions[55–59,61]. Offspring of genomes derived from meiocytes in which VRK-1 was depleted during the period of chromosome movements were enriched in duplications and deletions. This may be

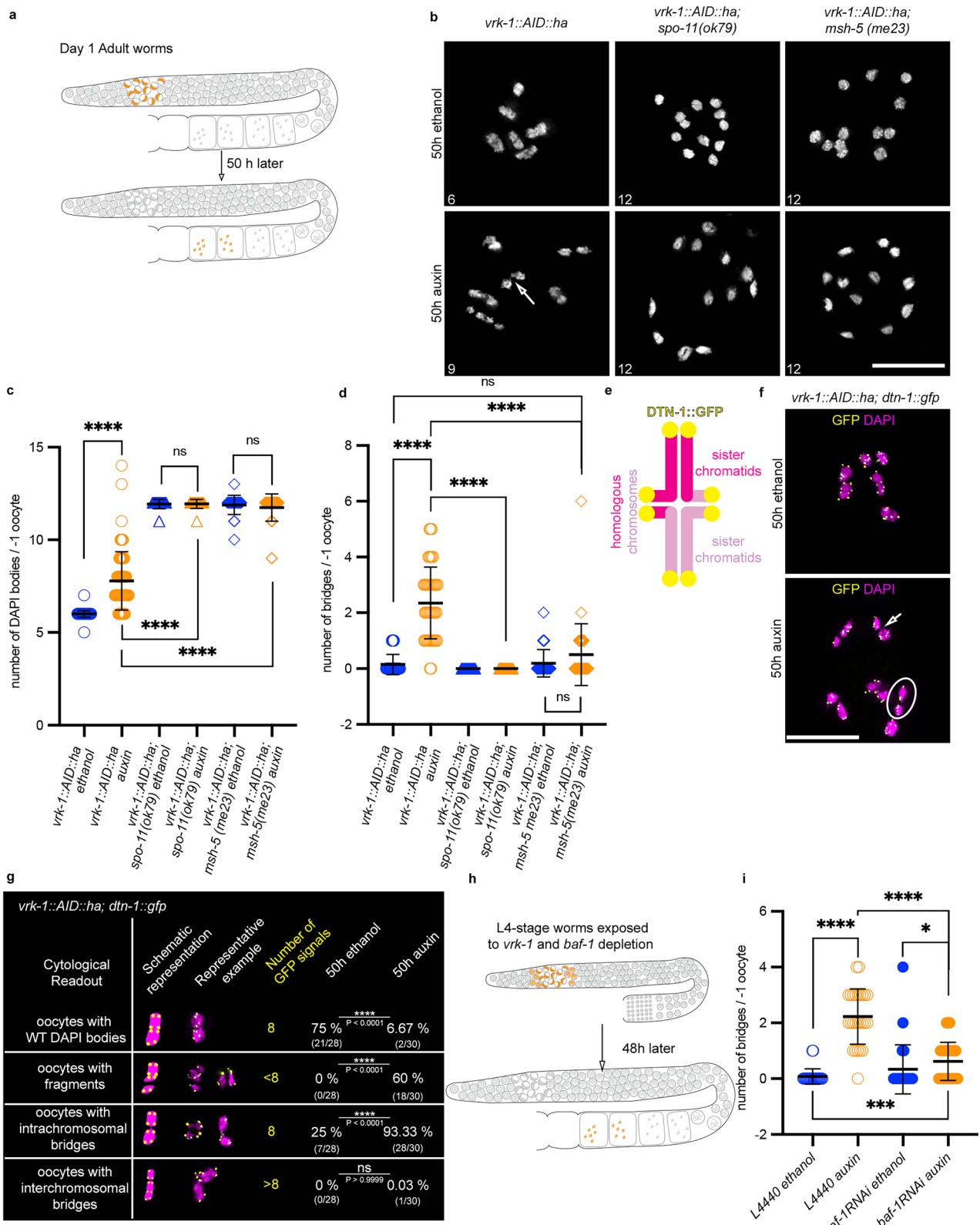

explained by ectopic recombination happening due to misalignment. Therefore, the timely release of chromatin from the nuclear periphery during the movement process helps to restrict ectopic recombination, which could otherwise cause the observed duplications and deletions.

Finally, VRK-1-depleted oocytes contain prominent SPO-11-dependent intrachromosomal bridges. Interestingly, these were observed after VRK-1 depletion at diplonema, when the recombination process is almost complete. This identifies an essential role for BAF-1 phosphorylation in this time window, analogous to the importance for NHK-1 for karyosome formation in *Drosophila*[19]. It is formally possible that the bridges are caused by less efficient chromatin condensation due to a lack of chromosome reorganization or by chromatin-accumulated BAF-1 interfering with the final resolution step of homologous recombination.

**Fig. 4 | VRK-1 depletion leads to the production of oocytes with aberrant chromosomes. a** Schematic representation showing the migration of transition zone nuclei after 50 h. **b** -1 oocytes stained with DAPI for the indicated genotypes and conditions. DAPI body numbers are indicated on the bottom left of each inset. Scale bar: 10 μm. **c** Quantification of DAPI bodies in the −1 oocyte for the indicated genotypes after 50 h treatment with ethanol or auxin. Mean ± SD per genotype (*n*= number of -1 oocytes): *vrk-1::AID::ha*, ethanol (*n* = 61): 6.0 ± 0.2, auxin (*n* = 70): 7.8 ± 1.6, *P* < 0.0001; *vrk-1::AID::ha; spo-11 (ok79)*, ethanol (*n* = 30): 11.9 ± 0.3, auxin (*n* = 33): 11.9 ± 0.2, *P* > 0.9999; *vrk-1::AID::ha; msh-5 (me23)*, ethanol (*n* = 26): 11.9 ± 0.5, auxin (*n* = 35): 11.7 ± 0.7, *P* = 0.5530; *vrk-1::AID::ha* auxin vs *vrk-1::AID::ha; spo-11 (ok79)* auxin and *vrk-1::AID::ha* auxin vs *vrk-1::AID::ha; msh-5 (me23)* auxin, *P* < 0.0001. The Two-sided Mann–Whitney test was used to assess statistical significance. **d** Number of chromosome bridges in the -1 oocyte for the indicated genotypes after 50 h treatment with ethanol or auxin. Mean ± SD per genotype (*n*= number of -1 oocytes): *vrk-1::AID::ha*, ethanol (*n* = 33): 0.2 ± 0.4, auxin (*n* = 46): 2.3 ± 1.3, *P* < 0.0001; *vrk-1::AID::ha; spo-11 (ok79)*, ethanol (*n* = 30): 0.0 ± 0.0, auxin (*n* = 33): 0.0 ± 0.0, *P* > 0.9999; and *vrk-1::AID::ha; msh-5 (me23)*, ethanol (*n* = 26):

0.2 ± 0.5, auxin (*n* = 35): 0.5 ± 1.1, *P* = 0.1761; *vrk-1::AID::ha* auxin vs *vrk-1::AID::ha; spo-11 (ok79)* auxin and *vrk-1::AID::ha* auxin vs *vrk-1::AID::ha; msh-5 (me23)* auxin, *P* < 0.0001. The Two-sided Mann–Whitney test was used to assess statistical significance. **e** Scheme depicting the localization of DTN-1::GFP. **f** DAPI (magenta) and GFP (yellow) stained -1 oocytes for the indicated genotype and conditions. Scale bar: 10 μm. **g** Schematic representations and representative examples of the different chromosome aberrations observed in *vrk-1::AID::ha; dtn-1::gfp* oocytes for the indicated conditions, showing numbers and calculated *P* values (Two-sided Fischer's exact test). **h** Scheme depicting the migration of transition nuclei from L4 stage in 48 hours. **i** Number of chromosome bridges in the -1 oocyte for the indicated genotypes and conditions. Mean ± SD per genotype (*n*= number of -1 oocytes): *vrk-1::AID::ha*: L4440−ethanol (*n* = 25): 0.1 ± 0.3 vs L4440−auxin (*n* = 26): 2.2 ± 1.0, *P* < 0.0001; *baf-1* RNAi−ethanol (*n* = 27): 0.3 ± 0.9 vs *baf-1* RNAi−auxin (*n* = 37): 0.6 ± 0.7, *P* = 0.0147; L4440−ethanol vs *baf-1* RNAi−auxin, *P* = 0.0004; and L4440−auxin vs *baf-1* RNAi−auxin, *P* < 0.0001. The Two-sided Mann–Whitney test was used to assess statistical significance.

De novo deletions and duplications have been reported in the male mouse germline, mainly in the absence of ATR (ataxia telangiectasia and Rad3-related protein) kinase, leading to an increased reliance on NHEJ[62]. Here, we identified a yet uncharacterized mechanism for generating large genome variants in an animal model that might be relevant to human disease.

In summary, our study sheds light on the role of the VRK-1–BAF-1 module in protecting against meiotic chromosomal abnormalities, a reduced oocyte pool, and the generation of genome variants such as duplications and deletions.

## Methods

### Nematode strains, strain construction, and culture conditions

All strains were derived from N2 Bristol unless otherwise stated and were cultivated under normal conditions. Strains were generated using the clustered interspaced short palindromic repeats (CRISPR)-Cas9 strategy following a published protocol[63], unless otherwise stated. The *vrk-1::AID::ha* strain was generated by Suny Biotech (https://www.sunybiotech.com) and then crossed with the strain expressing *tir-1::mRuby*. All CRISPR strains were outcrossed to N2 at least twice. For all depletion experiments using the AID system, worms were pre-picked at L4 stage and grown at 20 °C for 16–20 hours (h) overnight (O/N). The next day, they were transferred to nematode growth medium (NGM) plates containing ethanol (wild type) or 4 mM 3-indoleacetic acid (auxin; Sigma, cat. number I2886) and grown for 6, 18, 24, or 50 h. A list of the strains used in this study can be found in Supplementary Table 3. A list of crRNA, repair templates and genotyping primers generated in this study can be found in Supplementary Tables 4 and 5 respectively.

### Viability assays

L4 worms were pre-picked and incubated O/N at 20 °C and the next day were transferred to ethanol or auxin plates for the indicated time. Worms were transferred every day onto a fresh plate for 4 days. Eggs and number of hatched worms were counted. The percentage viability was calculated as the number of hatched eggs/total number of laid eggs) × 100. Unless specified otherwise, 10 worms per condition were used, and data are given as % mean viability ± SD.

### Cytological preparation and immunostaining of germlines

Germlines were prepared as previously described[64]. Briefly, gonads were dissected from young or older adult worms in 1× phosphate-buffered saline (PBS), fixed in 1% formaldehyde for 5 minutes (min) at room temperature (RT), and frozen in liquid nitrogen. After freeze-cracking, post-fixation in methanol for 1 min, and three washes in PBS containing 0.1% Tween-20 (0.1% PBST), non-specific binding sites were blocked using 1% bovine serum albumin (BSA, Sigma-Aldrich, cat.

number A9418) in 0.1% PBS-T for 1 hour at RT. Antibodies were diluted in 0.1% PBST and incubated with germlines O/N at 4 °C. The next day, germlines were washed three times in 0.1% PBST and incubated with secondary antibodies diluted in 0.1% PBST for 1.5 h at RT. After three washes in 0.1% PBST, samples were mounted in VECTASHIELD (Vector Laboratories, cat. number H-1000-10) containing 4′6-diamidino-2-phenylindole (DAPI; 2 mg/ml). A list of the primary and secondary antibodies used in this study can be found in Supplementary Table 6.

To visualize the DTN-1::GFP signal in diakinesis oocytes, the same procedure was followed but without primary and secondary antibody.

For visualization of MSH-5:: GFP, germlines were prepared as previously described[65]. Briefly, 10–15 young adult worms were dissected on poly-L-lysine-coated slides in 1× egg buffer containing 0.1% Tween-20 and immediately frozen in liquid nitrogen. After freeze-cracking, slides were incubated in methanol at −20 °C for 1 min. Gonads were then fixed with 4% paraformaldehyde in 100 mM $K_2HPO_4$ (pH 7.4) for 15 min, washed three times in 1× PBST and incubated with DAPI.

### EdU pulse labeling and migration rate analysis

EdU labeling was carried out as previously described[66–68] using the Click-iT™ Plus EdU Cell Proliferation Kit for Imaging with Alexa Fluor™ 594 dye (Thermo Fischer Scientific, C10639). Briefly, 30–40 synchronized young adult worms were incubated in 20 μM EdU in M9 buffer for 20 min at RT in the dark. Germlines were then dissected from 10 worms (t = 0), and the remainder were incubated on ethanol or auxin plates at 20 °C and germlines dissected after 18, 48 or 63 h. Dissected germlines were processed as described for the cytological preparation, DAPI was applied, and the Click-iT reaction was performed according to the manufacturer's instructions. VECTASHIELD was then applied and the slides were sealed and imaged.

### Chromosome spreads

Chromosome spreads were prepared as previously described[69] and analyzed using the LSM 900 Airyscan module. Briefly, 2-day-old hermaphrodites from two to three NGM plates containing ethanol or auxin were collected, and 20 μl from of the worm pellet was placed on a slide and chopped in dissection buffer (5% Dulbecco's Modified Eagle Medium (DMEM) in 0.1% Tween-20). Chopped worms (5 μl) were transferred to 24 × 40 mm coverslips and 50 μl of spreading solution was added (32 μl fixative (4% formaldehyde, 3.2% sucrose in water), 16 μl 1% lipsol in water, 2 μl 1% sarcosyl in water) and spread with a plastic tip. The coverslips were dried for O/N at RT and then washed in methanol at -20 °C for 20 min. Coverslips were then washed three times for 5 min each in 0.1% PBST and non-specific binding sites were blocked by incubating for 20 min in 1% BSA in PBST. Antibodies were

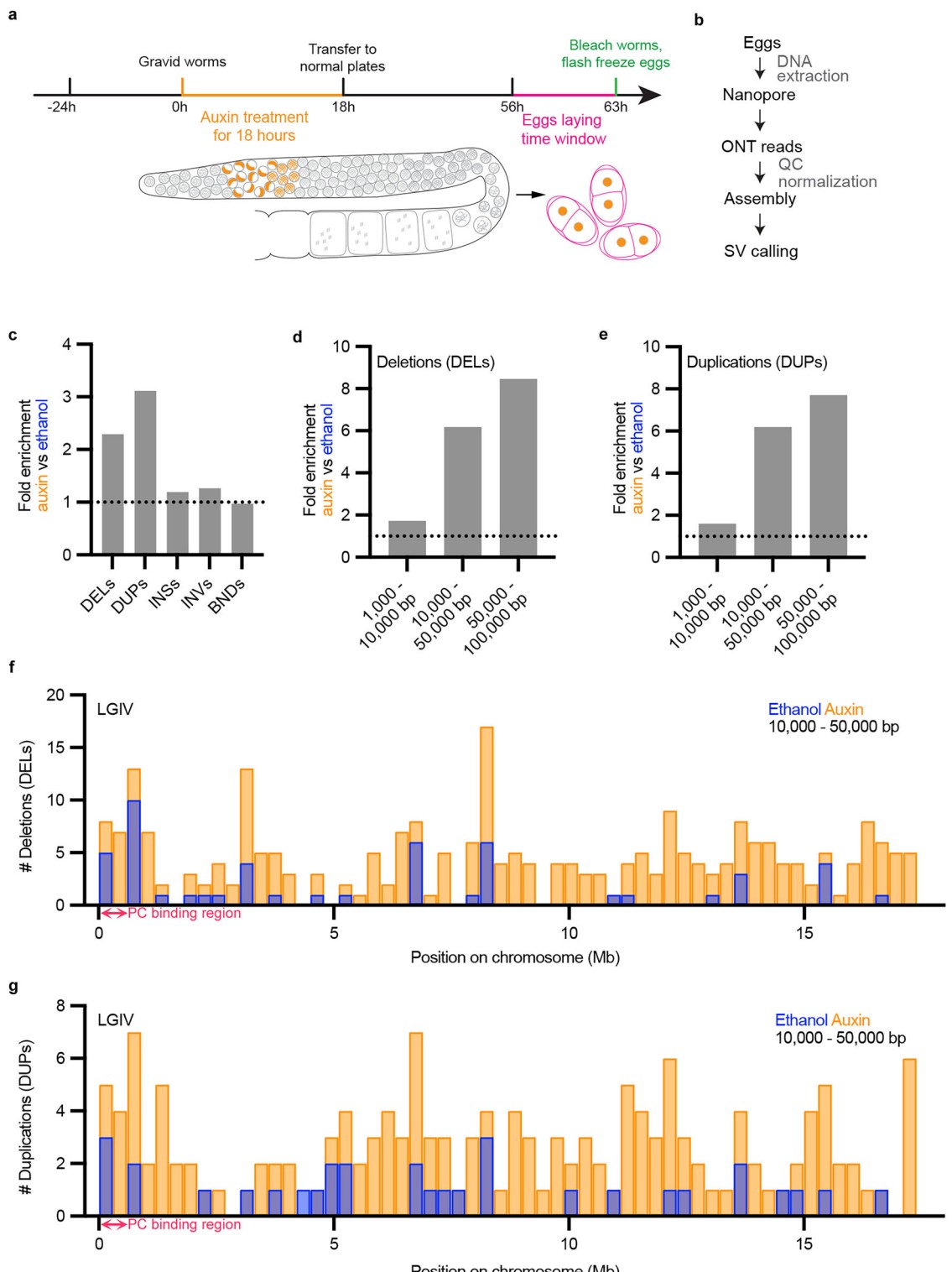

**Fig. 5 | VRK-1 depletion in leptotene, zygotene, and early pachytene meiocytes causes large deletions and duplications in embryos. a** Experimental outline for collecting embryos that were transiently depleted of VRK-1 in transition zone–early pachytene. **b** Schematic pipeline of the sequencing experiment and data processing. ONT = Oxford Nanopore Technology. **c** Fold enrichment of aberrant structural variants in auxin-treated samples. **d**, **e** Fold enrichment of deletions (**d**) and duplications (**e**) in various size bins. **f**, **g** Number of deletions (**f**) and duplications (**g**) of 10000–50000 bp along linkage groups IV for ethanol-treated (blue) and the auxin-treated (orange) germlines. The PC binding region is indicated (magenta). DEL = deletion, DUP = duplication, INS = insertion, INV = inversion; BND = translocation, LGIV = linkage group IV, PC = pairing center.

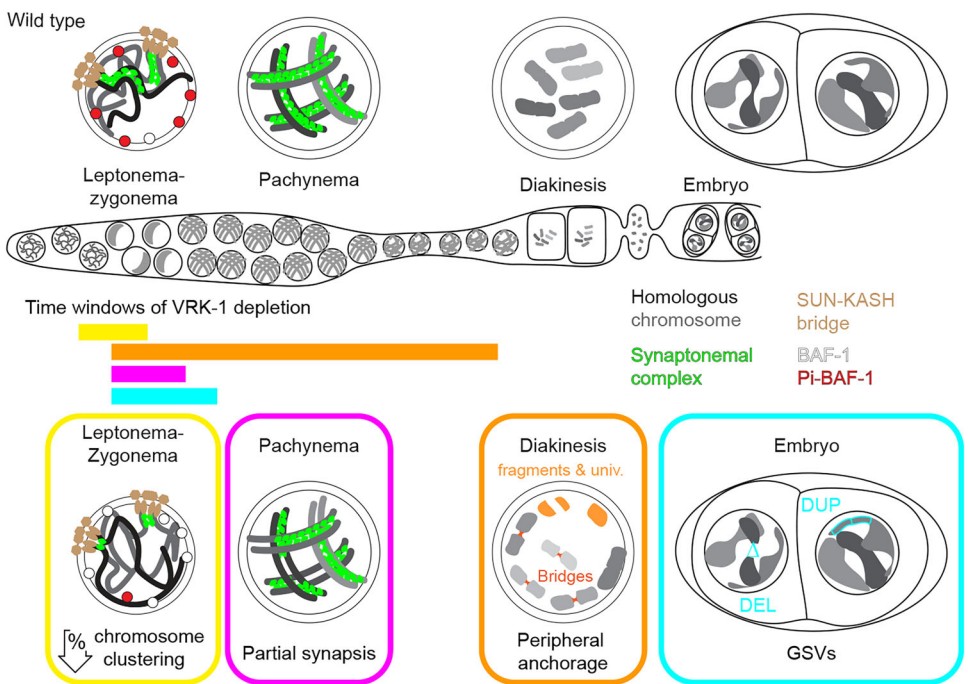

**Fig. 6 | Schematic representation of the roles of VRK−1 during meiotic prophase I in *C. elegans*.** Schematic insets show a comparison between the wild type (top) and VRK-1 depleted phenotypes (bottom), highlighting the lack of BAF-1 phosphorylation, partial synapsis, chromosomal aberrations, and genome structural variants (GSVs) in the offspring. univ = univalents.

diluted in blocking buffer and incubated O/N at 4 °C (primary antibodies) or 2 h at RT (secondary antibodies).

## Multiplex fluorescence in situ hybridization with oligopaints

Multiplex fluorescence in situ hybridization (FISH) was performed as previously described[34]. We received the library from the Kennedy lab and amplified probes for chromosomes I, II, and III according to the published protocol[34]. We then performed a second PCR to add a T7 polymerase site to the 5′-end of the produced amplicon, followed by T7 reaction to produce ssRNA. The ssRNA was reverse transcribed to ssDNA, residual ssRNA was removed by alkaline hydrolysis, and the long ssDNA oligos were purified using the DNA Clean & Concentrator-100 kit (Zymo Research, cat. number D4029). Probes were stored at 100 pmol/µl at 4 °C.

A synchronized population of 1-day-old young adults was incubated in ethanol or auxin for 6 h. Worms from two to three NGM plates were collected in M9 buffer, pelleted (3000 rpm for 30 sec), and washed twice more with M9 buffer. They were then resuspended in 10 ml M9 buffer, rocked for ~30 min at RT, pelleted, and aliquoted in 1.5 ml microcentrifuge tubes (30–50 µl packed worms per tube). Samples were placed in liquid nitrogen for 1 min and the frozen worm pellets were resuspended in cold 95% ethanol and vortexed for 30 sec. The samples were then rocked for 10 min at RT and sedimented (3000 rpm for 30 s). The pellet was washed twice in 1× PBST (containing 0.5% Triton X-100), 1 ml 4% paraformaldehyde solution (in 1× PBS) was added and then samples were rocked at RT for 5 min, washed twice with 1× PBST, and resuspended in 2× SSC (20× saline-sodium citrate (SSC) buffer (Thermo Fischer Scientific, cat. number 15557-044) diluted in water) for 5 min at RT. Samples were sedimented and resuspended in a 50% formamide, 2× SSC solution at RT for 5 min, and then incubated at 95 °C for 3 min followed by 60 °C for 20 min. Samples were sedimented and resuspended in 60 µl hybridization mixture (10% dextran sulfate, 2× SSC, 50% formamide, 100 pmol primary probe per chromosome, and 2 µl RNAse A (Qiagen, cat. number 19101)). Hybridization reactions were incubated in a 100 °C heat block for 5 min before O/N incubation at 37 °C. Samples were then washed with prewarmed 2× SSCT (2× SSC with 0.5% Triton X-100) with rotation at 60 °C for 5 min, followed by a second 2× SSCT wash at 60 °C for 20 min. Wash buffer was removed, and samples were resuspended in 60 µl bridge oligo hybridization mixture (2× SSC, 30% formamide, 100 pmol bridge oligo per targeted region, and 100 pmol each detection oligo). Bridge/detection oligo hybridization reactions were incubated at RT for 3 h. Samples were then washed in prewarmed 2× SSC at 60 °C for 20 min, followed by a 5 min wash with 2× SSCT at 60 °C and a 20 min wash in 2× SSCT at 60 °C. Samples were then washed at RT in 2× SSCT. Wash buffer was removed, and samples were resuspended in mounting medium (VECTASHIELD) containing DAPI. Samples were mounted on microscope slides and sealed with nail polish.

## FISH for the pairing analysis

The FISH protocol was based on a published protocol[70]. Dissected gonads were fixed in 4% paraformaldehyde in egg buffer for 2 min at RT and then stored in methanol at -20 °C. Slides were then incubated in methanol at RT for 20 min, followed by 1-min washes in 50% methanol and 1× SSC buffer containing 0.1% Tween-20 (SCCT) and dehydration by sequential immersion in 70%, 90%, and 100% ethanol (3 min each). Hybridization mixture containing 10.5 µl FISH buffer (1 ml 20× SCCT, 5 ml formamide, 1 g dextran sulfate, 4 ml H$_2$O), and 2.5 µl labeled probe was added to air-dried slides. The FISH probe for the 5S rDNA locus (chromosome V) was made by labeling 1 mg DNA with the DIG (Digoxigenin)-nick translation kit (Sigma-Aldrich, cat. number 11745816910). After the addition of ethylenediaminetetraacetic acid (EDTA), the probe was incubated at 65 °C for 10 min. PCR-amplified 5S rDNA was used to probe the right end of chromosome V and was labeled by PCR with digoxigenin-11-deoxyuridine triphosphate. Slides were incubated at 37 °C O/N in a humidified chamber and then washed twice (20 min) at 37 °C in the following buffer: 50% formamide, 2× SCCT, and 10% Tween-20. After three washes in 2× SCCT at RT, samples were blocked for 1 hour in 2× SCCT containing 1% BSA (w/v). Slides were then incubated in secondary anti-biotin antibody diluted in 2× SCCT (1:500) for 2 h at RT, followed by three washes in 2× SCCT, and then stained with DAPI (1 ng/ml) and mounted in VECTASHIELD.

## RNA interference

RNAi was performed as previously described[42]. Briefly, a single colony from the *baf-1* clone and from the empty vector L4440 (Ahringer collection[71]) were grown O/N at 37 °C in 2× Tryptone-Yeast (TY) medium supplemented with ampicillin (100 μg/ml). The next day, cells were pelleted at 3000 *g* for 15 min and resuspended in 2× TY. Cultures were split into two and supplemented with ethanol or auxin to a final concentration of 4 mM (resulting in four different cultures: L4440–ethanol, L4440–auxin, *baf-1*–ethanol, and *baf-1*–auxin). Each suspension (150 μl) was used to seed NGM plates containing 1 M IPTG (isopropyl-β-d-thiogalactopyranoside) and ampicillin (100 μg/ml) and incubated at 37 °C O/N. Pre-picked L4 were added to the plates and grown at 20 °C for 48 h before analysis.

## Nuclei isolation and protein fractionation from large auxin-treated and control *C. elegans* cultures

Nuclei isolation and cellular fractionation were done as previously described[72]. Large cultures of *C. elegans* were prepared by seeding 50 100-mm NGM plates with 1 ml OP50 bacteria (obtained by centrifugation of 3 l of O/N *E. coli* culture and resuspending the pellet into a final volume of 50 ml). Another 50 100-mm NGM plates were also seeded with 1 ml concentrated OP50 bacteria supplemented with ethanol or auxin. Synchronized population of *vrk-1::AID::ha* L1 worms were grown at 20 °C to 1-day-old adults on the 100-mm plates containing OP50 bacteria. Young adult worms were collected and transferred to NGM plates supplemented with ethanol or auxin and grown for another 18 h. Worms were then collected in 50 ml tubes by washing the plates with M9 buffer, pelleted by gravity, and washed three times with M9 buffer. After the last wash, excess buffer was removed and the worm pellet was split into 15 ml tubes, each containing 1 ml packed worms and 3 ml of NP buffer (10 mM HEPES-KOH (pH 7.6), 1 mM EGTA, 10 mM KCl, 1.5 mM MgCl$_2$, 0.25 mM sucrose, 1 mM phenylmethylsulfonyl fluoride containing protease inhibitors (Roche, catalog no. 11836170001)). Nuclei were isolated from one tube each of auxin-treated and control worms. For this, worms were disrupted using a cooled metal Wheaton tissue grinder and the suspension was filtered with a 100 μm mesh and then with a 40 μm mesh. The filtered solution was centrifuged at 300 *g* for 2 min at 4 °C, and the supernatant (containing nuclei) was centrifuged at 2500 *g* for 10 min at 4 °C. The pellet contained germline nuclei and the supernatant was used as the cytosolic fraction #1. Cytosolic fraction #2, soluble and DNA-bound protein nuclear (nuclear insoluble) fractions were prepared using a Qproteome Nuclear Protein Kit (QIAGEN, cat. number 37582) according to the manufacturer's instructions.

## Whole-worm extracts

Worm samples were prepared as previously described[70]. Preselected L4 worms (200 per genotype per assay) were incubated at 20 °C for 24 hours. Adults were then collected into 30 μl of TE buffer [10 mM tris and 1 mM EDTA (pH 8.0)] in a 1.5-ml Eppendorf tube. After the addition of 5× Laemmli buffer, worms were subjected to three cycles of freeze thawing. Western blot for the samples was performed as described in the following section.

## Western blotting

Worm samples were prepared as previously described[70]: the four fractions obtained from the cellular fractionation (cytosol #1, cytosol #2, nuclear soluble, nuclear insoluble- 50 μg each for the GAPDH, HA, FLAG and 1 μg for the H3) were mixed with Laemmli buffer. Samples were separated by electrophoresis in 1× SDS-tris-glycine buffer on precast 4–20% TGX™ gels (Bio-Rad, cat. number 4561096EDU). Proteins were transferred onto polyvinylidene difluoride membranes (activated in methanol for 20 sec) for 1 hour at 4 °C at 100 V in 1× tris-glycine buffer containing 20% methanol. Membranes were blocked for 1 hour in 5% milk dissolved in 1× tris-buffered saline containing 0.1%

Tween-20 (TBST); primary antibodies were added to the same buffer and incubated O/N at 4 °C. Membranes were then washed three times in 1× TBST and incubated with the secondary antibody in 5% milk in 0.1% TBST for 1 hour at RT. After three washes, membranes were incubated with WesternBright Enhanced Chemiluminescence (ECL) substrate (Advansta, cat. number K-12045-D20) and developed using a ChemiDoc system (Bio-Rad). Primary and secondary antibodies are listed in the Supplementary methods. Stripping of membranes (when required) was performed using the Restore™ Western Blot Stripping Buffer (Thermo Scientific, cat. Number 21059) according to the manufacturer's instructions.

## Generation of phospho-antibody against BAF-1 (S4Pi)

The polyclonal antibody was generated by Eurogentec (https://www.eurogentec.com/) and the peptides used for raising the anitbodies against BAF-1 (S4Pi) and BAF-1 were H-MST S(PO3H2)VKHREFVGC - NH2 (13AA) and H-MSTSVKHREFVGC - NH2 (13AA) respectively (where H at the N-terminus means free amine (NH2-) and NH2 at the C-terminus means amide (-CONH2)).

## SYTO12 staining

SYTO12 staining was performed as previously described[70]. One-day-old adult worms were incubated in ethanol or auxin for 18 h and then soaked in 33 μM SYTO12 (Thermo Fisher Scientific, cat. number S7574) in M9 buffer for 2–3 h at 20 °C in the dark. Worms were then transferred to unseeded NGM plates for 30 min, and then mounted on slides. SYTO12-positive cells within the germline were scored using an epifluorescence microscope equipped with a 40× oil immersion objective lens.

## Live Imaging of worms for monitoring chromosome movements

For analyzing the movement of SUN-1::GFP, worms were treated as previously described[7] with some modifications. Adult hermaphrodites were pre-selected at the L4 stage and incubated in ethanol or auxin as day 1 young adults for 18 h before being mounted onto 2% agarose pads in M9 buffer containing 1 mM tetramisole for filming. Coverslips were sealed with melted petroleum jelly. Images were acquired as 0.8-mm thick optical sections every 5 s for 5 min. Data were analyzed using ImageJ 1.54 h (NIH) with StackReg and Manual Tracking plugins. Aggregate tracking was performed with Rstudio 2025.05.01 + 513.

## Preparation of samples for Oxford Nanopore Technology sequencing

A synchronous population of *vrk-1::AID::ha* worms (through bleaching) were allowed to develop to the 1-day young adult stage. Worms were then collected in M9 buffer and washed twice. The worm pellet was divided between four ethanol and four auxin plates (4 mM) and incubated for 18 h. Worms were collected in M9 buffer, transferred to normal NGM plates, and allowed to recover for 48 h. On the 3rd day post recovery, worms were transferred to fresh plates. After 8 h, they were bleached so that only eggs from this time window could be recovered for sequencing.

Genomic DNA was extracted from approximately 40,000 ethanol-treated and 30,000 auxin-treated eggs following a protocol adapted from the Puregene Core Kit A for tissue (Qiagen, cat. number 158063). To digest the eggshell, 60 μl of 20 mg/ml chitinase in egg buffer (pH 6.5) was added to the frozen egg pellet and resuspended by pipetting. Digestion was monitored under a stereoscope. Additional chitinase (10 μl) was added after 30 min and again after another 10 min to maintain enzyme activity. Digestion was carried out for approximately 1 hour, or until the embryos appeared flattened. The reaction was then stopped by flash freezing and samples were stored at -20 °C. For DNA extraction, the worm pellet was transferred to a 1.5 ml tube, 600 μl Cell Lysis Solution and 5 μl Proteinase K were added, and mixed by inverting the tube 25 times. Samples were incubated at 55 °C for 3–4 h,

with gentle inversion every hour until the solution became clear. To remove RNA contamination, 3 μl RNase A was added and mixed by inversion 25 times, followed by incubation at 37 °C for 1 hour, with mixing every 15 min, followed by incubation on ice for 1 min.

To precipitate proteins, 200 μl Protein Precipitation Solution was added and samples were vortexed vigorously for 20 sec before incubation on ice for 5 min and centrifugation at 16,200 g for 3 min at 4 °C. The supernatant was transferred to a clean 1.5 ml tube containing 600 μl isopropanol, 0.5 μl glycogen solution was added to enhance DNA precipitation and inverted 50 times to mix. After centrifugation at 16,200 g for 15 min at 4 °C, the supernatant was carefully discarded and the pellet was air-dried on absorbent paper.

The DNA pellet was washed three times with 70% ethanol, followed by centrifugation at 16,200 g for 15 min at 4 °C. After the final wash, the ethanol was carefully removed, and the DNA was left to air dry completely. The pellet was then resuspended in 100 μl DNA Hydration Solution and incubated at RT O/N. The DNA concentration was measured using a Qubit 2.0 fluorometer (Thermo Fischer Scientific), and DNA integrity was assessed by 0.7% agarose gel electrophoresis. DNA samples were then submitted to the Vienna BioCenter facility (https://www.viennabiocenter.org/vbcf/) for analysis.

### Microscopy and evaluation

Microscopy images were acquired as previously described[70]. Briefly, three-dimensional (3D) image stacks were obtained using a DeltaVision Ultra High-Resolution microscope equipped with 100×/1.40 oil immersion objective lenses and a custom softWoRx software package. Acquired images were deconvolved using the softWoRx deconvolution algorithm. Maximum intensity projections of deconvolved images were generated using ImageJ after adjusting the maximums and with background subtraction using a rolling ball radius of 50 pixels. Where specified, images of gonads consist of multiple stitched images. This was necessary because of the size limitation of the field of view at high magnifications. Image stitching to build up entire gonads was performed manually in Adobe Photoshop. Fluorescence levels across stitched images were adjusted in Adobe Photoshop to correct for auto-adjustment by the microscope.

Chromosome spreads and multiplex FISH-oligopaint preparations were imaged using an LSM 980 (100×/1.46 objective lens) and LSM 900 (63×/1.46 objective lens), respectively, and an Airy scan module.

For evaluation, processing, and quantification of intensities of the acquired images, see the Supplementary methods.

### Statistics and Reproducibility

Datasets were plotted and statistical analyses performed in GraphPad Prism 10.4.0 for Mac. Datasets were tested for normal distribution and tested for significant differences using the two-tailed Fisher's exact test, two-sided Mann–Whitney test, two-sided Two-Proportion Z-Test, two-sided Kruskal–Wallis test, multiple two-sided Mann–Whitney tests or two-sided Chi-square test, as appropriate. $P$-values were adjusted in case of multiple comparisons as appropriate. The exact number of $P$-value has been provided where possible. For Figs. 1c, m, 2b, g, 3c, e, 4c, d, g, i the exact values were not possible to be provided since the softare calculated a $P$-value < 0.0001 or >0.999. Effect size was tested using the Cliff's delta test in Python.

The micrographs presented (Figs. 1d, e, 2c, d, 4f) have been derived from at least two independent experimental datasets (more than 5-6 micrographs can be obtained per independent experiement). The fractionation and the western blot (Fig. 2h) have been repeated in two biological replicates. Detection using the BAF-1(S4Pi) antibody has been repeated once.

### Data availability

All obtained sequencing reads generated in this study have been deposited in the NCBI public database and are available under the project PRJNA1304376. All remaining data generated or analyzed in this study are included in this published article and its Supplementary information files. Source data are provided with this paper.

### Code availability

The VCF files generated by Sniffles and cuteSV were parsed and analyzed using the Python scripts/Jupyter notebooks provided on the GitHub repository at https://github.com/Melkrewi/Structural_variants_detection[73]. SV breakpoints were analyzed for the presence of short tandem repeats (STRs) with unit sizes of 1, 2, or 3 bp using STR-Finder provided in GitHub repository at https://github.com/Single-Molecule-Genetics/STR-Finder[74].

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

## Acknowledgements
We are grateful to Monique Zetka, Nicola Silva, and Yumi Kim, Needhi Bhalla, George Krohne and Rueyling Lin for providing reagents; Scott Kennedy for sharing the multiplexed FISH library; and members of the Max Perutz Labs' BioOptics facility (Irmgard Fischer, Josef Gotzmann, Thomas Peterbauer, Clara Bodner, and Nick Wedige) for training and support in image acquisition. We also thank the members of the NGS facility at the Vienna Biocenter. This work was funded by the Austrian Science Fund (FWF) SFB projects F 8805-B (VJ), https://doi.org/10.55776/F88, F 8809-B (ITB), and F8810-B (BV). We are also grateful to members of the V. Jantsch laboratory for helpful discussions. Some strains were provided by the *Caenorhabditis* Genetics Center, which is funded by the National Institutes of Health Office of Research Infrastructure Programs (P40OD010440).

## Author contributions
D.P. performed all experiments unless otherwise stated. A.B. performed the multiplexed FISH and multiplexed FISH image analysis, 3D diakinesis analysis, and sample preparation for the ONT and prepared the figures with the sequencing data after R.K. and M.E. provided the relevant scripts. R.K., M.E., and B. V. performed the sequencing analysis of the ONT reads. S.M.A.A. and I.T.-B. analyzed the repeat sequences from the ONT reads. A.G. contributed to the construction of *C. elegans* strains. C.K. acquired the images of *baf-1* mutants and analyzed mitotic length in the BAF-1 phospho-mutants. The *vrk-1::AID::ha* strain was constructed by Suny Biotech (https://www.sunybiotech.com). D.P., A.B., and V.J. conceived all experiments; D.P. and V.J. wrote the manuscript, with input from A.B., R.K., M.E., I.T-.B. and B.V; and V.J. acquired the funding.

## Competing interests
The authors declare no competing interests.
