## [Transparent Peer Review file · Nature Communications]

BAF-1–VRK-1 mediated release of meiotic chromosomes from the nuclear periphery is important for genome integrity

Corresponding Author: Professor Verena Jantsch

Version 0:

Reviewer comments:

Reviewer #1

(Remarks to the Author)

In this work by Paouneskou et al., the authors addressed a long-standing question in meiosis – the mechanisms and roles of regulated chromosome tethering at the nuclear periphery during meiotic prophase – by combining induced protein degradation, chromosome painting, cytology, mutant/genetic analyses, and long-read sequencing. Meiosis depends on rapid chromosome movement in early prophase to achieve homolog pairing, synapsis, and eventually accurate segregation. In *C. elegans* – a premier experimental model system for studying meiosis – it's well established that the reorganization of chromosomes into a "crescent shaped" mass coincides with their rapid movement across the nuclear envelope (NE) in early prophase. However, the relationship between chromosome reorganization and their rapid movement remained unclear. Specifically, it remained enigmatic whether rapid chromosome movement was sufficient to "untether" chromosome from the NE, or additional mechanism might be required for this process. By focusing on a kinase (VRK-1) and the chromatin-NE tethering protein BAF-1, the authors identified a new mechanism that mediates the regulated tethering between chromosomes and the NE in early prophase. The authors first used auxin-inducible degradation to deplete VRK-1 and demonstrated that it caused "overtethering" of chromosomes in both leptotene/zygotene and diakinesis nuclei. They also demonstrated that the "overtethering" phenotype can be recapitulated by specific nonphosphorylatable mutations in BAF-1. They then carefully documented a series of meiotic phenotypes including delayed homolog pairing, synapsis, crossover formation, as well as elevated apoptosis upon VRK-1 depletion. Additionally, the authors showed that VRK-1 depletion led to aberrant chromosomes in maturing oocytes, in a manner that is dependent on BAF-1 and meiotic recombination. Finally, by taking advantage of the reversibility of the auxin inducible degradation system, the authors carried out an impressively meticulous assay to deplete VRK-1 only during transition zone and early pachynema and observed increased deletions and duplications in the next generation embryos. Overall this work was very well executed, with all the experiments well controlled. This work elucidated how VRK-1/BAF-1 regulates the tethering of chromosomes at the NE during meiosis, likely independent of rapid chromosome movement, with far-reaching implications for the roles of chromosome reorganization at the nuclear periphery in genome stability. Specific comments/ points are as follows. It should be noted that the points raised below did not dampen the high enthusiasm for this work and the perceived overall high interest and broad impact it will generate in the fields of meiosis, genetics, and cell biology of the nucleus.

(1) It would be helpful to clarify the localization of VRK-1, especially during leptotene/zygotene, where it was implicated to phosphorylate BAF-1 to "untether" chromosome from nuclear periphery. Do VRK-1 have more obvious NE localization in leptotene/zygotene nuclei compared to pachytene nuclei? From Supp Fig. 1c and Supp Fig. 5b, VRK-1 seem to have foci-like localization at the NE, however, this is obscured by the SUN-1(S8)Pi staining in the merged images in Supp Fig. 1c. It'd be informative to include monochromatic images and intensity measurement of VRK-1 in relation to an NE marker and/or chromosomes.

(2) The idea that chromosome movement is not affected by VRK-1 depletion (or BAF-1 phosphorylation) can be very interesting, because this could present a unique case where the "crescent shaped" chromosome organization is uncoupled from chromosome mobility during early meiosis. Have the authors looked at whether SUN-1 patches colocalize with chromosome pairing centers, especially upon VRK-1 depletion? This would be important to the conclusion that chromosome mobility can be uncoupled from VRK-1-mediated untethering from nuclear periphery. On a related note – have the author examined whether the length of SUN-1-Ser8Pi positive region change upon VRK-1 depletion?

(3) Per Supp Fig. 4f & 4g, it seems that the intrachromosome bridge phenotype can be caused by late/short depletion of

VRK-1 alone (even from diplonema), whereas increased numbers of DAPI bodies require much longer depletion of VRK-1 (from transition zone). This suggests that intrachromosome bridge could be a separate phenotype specific to VRK-1/BAF-1's role during late prophase. Interestingly, VRK-1 depletion also caused "overtethering" of chromosomes to the NE in diakinesis oocytes (Fig. 1j, where it says "18hr auxin" in the figure panel; yet it's "50 hr" in the main text). Have the authors examined whether chromosome overtethering to the NE in "-1" oocytes can also happen with short auxin treatment (i.e. VRK-1 depletion from diplonema) that results in the bridge phenotype?

(4) The authors probably already have the data – how does *flag::baf-1-S4A* affect chromosome anchorage, bivalent number, or intrachromosome bridge in diakinesis oocytes?

Minor points:

(1) It'd be helpful if the authors can clarify whether VRK-1 depletion in early meiosis, which caused increased deletions and duplications in the offspring, had any functional consequences. It was noted that after the 18 hr VRK-1 depletion, "embryo viability were similar to the wt following VRK-1 resynthesis (Supp Fig. 5b)". Yet in Supp Fig. 5b, embryo viability seems to be indeed significantly lower than wt 2 days after VRK-1 depletion for 18hr, which appears to be the targeted time window according to Fig. 5a. If this is correct, the authors might consider highlighting it to underscore the physiological importance of VRK-1/BAF-1 regulated chromosome tethering. Also, have the authors checked male indices?

(2) In their introductory section discussing lamin's role in conferring "stability to germline nuclei", the authors may consider referencing to additional work such as [PMID: 37436986], [PMID: 21529718], and [PMID: 11071918], which could provide a fuller context and strengthen the scholarly background.

(3) The numbers in Supp Table 1 have a mixture of units (raw counts versus percentages), making comparisons (e.g. between different *baf-1* mutants) confusing.

(4) In Acknowledgements and Table 4, Rueyling Lin's name was misspelled as Rueyling Kin.

Reviewer #2

(Remarks to the Author)

The manuscript by Paouneskou and coworkers reports that acute depletion of Vaccinia Related Kinase 1 (VRK1) causes multiple meiotic defects in the nematode *Caenorhabditis elegans*. VRK1 is essential and to control its expression, the protein is fused endogenously with a degron tag that is recognised by TIR1 in the presence of auxin, which triggers ubiquitination and degradation. By multiplexed FISH the authors demonstrate that chromosome organization is abnormal already within 6 hours of exposure to auxin and chromosomes fail to be released from the nuclear periphery of diakinesis oocytes after 18h. Barrier to Autointegration Factor 1 (BAF1) is a known VRK1 substrate and by mutation of two phosphorylation sites in BAF1, the authors demonstrate that the role of VRK1 in meiosis is at least partially through regulation of BAF1. Extended VRK1 depletion (by 50h auxin treatment) results in the formation of aberrant chromosomes, including fragmented chromosomes and intrachromosomal bridges. Finally, the authors show that a 18h pulse of VRK1 depletion followed by approximately 48h recovery leads to the formation of eggs with a higher frequency of chromosomal deletions and duplications. Altogether, the data represent a novel layer of regulation of chromosome behaviour during animal meiosis regulated by the protein kinase VRK1 and its substrate BAF1 at the nuclear envelope.

Technically, the study is very impressive and overall convincing. The quality of the microscopy images is impressive, and the manuscript is well-written. Nevertheless, the authors are recommended to consider the following points prior to publication:

1) The discrimination between clustered and non-clustered chromatin (e.g. Figures 1c and 2g) seems to be based on visual inspection. Because this is a subjective evaluation, the evaluator should not know the genotypes during scoring. Was this the case?

2) Page 5. Based on Figure 1e, the authors state that depletion of VRK1 causes increased association of chromosomes to the nuclear periphery. However, this should be quantified (or wait with the claim until describing the data in Figure 1j-l).

3) According to Figure 1l, depletion of VRK1 causes chromosome bivalents to accumulate at the nuclear periphery in diakinesis nuclei. According to the main text, the experiment was performed by exposure to auxin for 50 h but the figure indicates 18h. Please clarify. Moreover, have shorter time points been tested? Fig S1c indicate that VRK-1 is efficiently depleted from the pachynema zone after only 6 h: from this zone the chromosomes condense shortly after.

4) On page 7 the authors claim that "Depletion of both *baf-1* and VRK-1 also released tethered chromosomes from the nuclear periphery in the -1 diakinesis oocyte (Fig. 2c), whereas *baf-1* depletion alone did not." This should be quantified (only a single image is shown).

5) The data in Figure 2d-e indicate an unexpected increase in nuclear BAF1 signal upon VRK1 depletion. How can this increase be explained? BAF1 is usually a nuclear protein so changes in nucleocytoplasmic transport is unlikely to be the explanation. Change in transcription and/or protein stability? Is this also observed more acutely, for instance after 6 h of exposure to auxin?

6) On page 7 the authors argue that "Chromatin reorganization was impaired in both [*baf1*] mutants". However, the T3A

mutant seems to behave as the wild type in all assays. Related to this, did the authors consider to simultaneously mutate both residues?

7) The interpretation of the Western blot depicted in Figure 2h is not obvious. Firstly, there seems to be less material in the nuclear fractions in the auxin sample compared to the ethanol sample (e.g. less H3 and FLAG signals), which may contribute to the already weak purple band indicated with the left white arrow to become invisible (right white arrow). Secondly, multiple purple bands seem not to be recognised by the FLAG antibody, which is unexpected. Finally, from the Methods section, it is not clear if this analysis was performed multiple times, i.e. if it is reproducible. Information on how the antibody was generated is also missing. For instance, will it recognise BAF phosphorylated both on T3 and S4 or only when phosphorylated exclusively on S4?

8) Regarding the Nanopore sequencing: The authors state that "insertions and inversions seem to be evenly distributed along the genome". In worms, attachment to the nuclear periphery is mainly through the distal ends of chromosomes. If the role of VRK1 is through phosphorylation of BAF1 and release of chromosome (distal ends), perhaps one could expect to see an increase in rearrangements in these regions? Or do the authors propose that the location of lesions is independent of distance to the anchoring to the nuclear periphery through BAF1?

9) Can the authors (roughly) estimate how many deletions and duplications were observed per cell or per embryo? The VRK1-depleted embryos have more mutations but normal viability according to the text, which seems surprising.

10) General point: many graphs report multiple statistical analyses within the same graph and experiment, but it is not clear if p values were adjusted accordingly. Some p values are borderline to be considered significant and may be affected by such adjustment.

Minor points:

Page 4. The sentence "vrk-1(syb2608 vrk-1::AID::ha); tir-1::mRuby (hereafter called vrk-1::AID::ha)" is incomplete and needs revision. The authors should mention in which tissue and/or when during development TIR1 is expressed.

It would be relevant to show examples of microscope images of EdU pulse labelling to complement the cartoons.

Chromosomes I, II and III were analysed by multiplexed FISH. Why were these chromosomes analysed and not IV, V and X?

What does the arrowhead in Figure 3b indicate?

Please revise the percentages in Figure 4e: $1/30 = 3.3\%$ (not 0.03%); $18/30 = 60\%$ (not 62.5%).

Figure 4f-g: Why was this done for 48h (vs 50h for the rest of Fig4) and from L4 (vs from day 1 adult for the rest of Fig 4)?

Regarding the apoptosis analysis, the authors should comment that this observation is opposite to the situation in a vrk-1 loss of function allele (Waters et al, 2010).

Supplementary Table 1: The left column reports (partial) genotypes but not strain names as the title indicates.

Reviewer #3

(Remarks to the Author)

Reviewer #4

(Remarks to the Author)

The authors of this mostly well-executed and rigorous study exploits the auxin induced degradation of VRK-1, an essential protein, during meiotic prophase. Using acute depletion of VRK-1, the authors show that a characterized VRK-1/BAF-1 axis that ensures that chromosomes are removed from the nuclear periphery with entry into mitosis plays a similar essential role during meiosis, actively regulating the scheduled removal of chromosomes from the nuclear periphery during entry into meiotic prophase. Thus, this work demonstrates that this removal is an active process that is not merely accomplished by the chromosome mobility that is a conserved aspect of meiotic prophase, a novel finding. They show that this dispersal does not have consequences on chromosome movement but does have consequences on pairing, synapsis and recombination. Moreover they show that this axis takes advantage of a conserved phosphorylation site on BAF-1, serine 4. The most impactful part of the paper, aside from the identification of this mechanism, is the authors' demonstration that the inability to remove chromosomes from the nuclear periphery upon entry into meiotic prophase has dramatic consequences for genomic stability, where deletions, duplications, insertions and inversions are increased when this removal is prevented by acute depletion of VRK-1. This result shows that this regulated dispersal likely limits ectopic recombination and ensures that meiotic recombination and crossover formation occurs correctly, maintaining genome stability during sexual reproduction. I do not have the expertise to comment on the sequencing approach. However, I have a few major concerns about the other

experiments, mostly controls, that need to be addressed.

Major points

1. Throughout the paper, the authors use the term chromatin organization. Given the role that tethering may play in regulating gene expression in interphase, I understand this choice. But using the term chromatin organization raises questions about histone modifications, transcriptional activity, etc. I suggest the authors use the term chromosome organization instead to describe the changes they see upon VRK-1 depletion.

2. In general, the authors could make their last point, about the increase in structural chromosomes rearrangements when VRK-1 is depleted in meiosis, stronger. That they can infer this from their cytological analysis and show it with their sequencing analysis is powerful. This work makes a compelling argument that this mechanism helps reorganize chromosomes in the nucleus upon entry into meiotic prophase and has profound consequences on genome stability, especially since chromosome mobility at PC ends seems normal. Given previous data that defects in bouquet formation increases ectopic recombination in yeast (Goldman and Lichten, 2000), this result supports a model where chromosomes have to be removed from the nuclear periphery to facilitate pairing, synapsis and accurate meiotic recombination and maintain genomic stability during a dangerous but necessary event, meiotic recombination.

3. The authors could also explain details about the *C. elegans* germline better to make the study more accessible to people in other fields. For example, the authors should point out the spatio-temporal organization of meiotic nuclei, especially with respect to their analysis of synapsis and recombination in Figure 3. Similarly, the authors could explain better that the connected bivalents are the products of recombination in their analysis in Figure 4.

4. The section on biochemical experiments needs to be expanded, to explain why and how this fractionation technique was undertaken and additional experiments are necessary, including important controls. Moreover, I could not find a description of the methods used to generate the phospho-specific antibody.

- What happens when they use the anti-phospho Ser4 antibody on germlines or lysates from baf-1S4A mutants? This is a necessary control to ensure the specificity of the antibody, especially given the signal that is still visible in Figure 2h in auxin-treated worms (see below).

- Although I agree that there I can't detect phosphorylated Baf-1 in the nuclear insoluble fraction in in auxin-treated VRK-1::AID::HA worms, why is there still phosphorylated Baf-1 in the cytosolic and nuclear soluble fractions? Is there another kinase that phosphorylates Baf-1 in these fractions?

- Shouldn't there be more Baf-1 associated with chromatin in auxin-treated VRK-1::AID::HA strains, especially given their immunofluorescence data in Figure 2d and e? If the authors think this discrepancy is because only specific regions of the germline exhibit this enrichment upon treatment with auxin, they should explicitly say that.

- Can the authors provide the methods for generating anti-phospho Ser4 antibody?

5. Are the authors surprised to see the number of structural variants in their ethanol controls? Does this say something about the functionality of the VRK-1::AID::HA transgene, even if it supports viability? Or does it say something about the mutagenic potential of meiotic recombination? Would you see similar numbers of structural variants in wildtype animals? This experiment is not strictly required but some commentary would be useful.

Minor:

1. The authors should spell out wildtype instead of using "wt."

2. Can the authors use a different color scheme for some Figures that make images easier to see, such as Figure 2D? The yellow and magenta on white is particularly difficult to see.

3. The authors try to tie their results back to what has been seen in other systems as a way to showcase conservation, but those examples could be written to be more clear and more explicit in their connections.

4. Can the authors indicate the Pairing Center end of chromosomes when reporting on the frequency of structural variants (Figures 5F and G and Supplementary Figures 9-20)?

5. The authors should check their spelling. For example, Needhi Bhalla is spelled incorrectly (Nheedi Bhalla)

Version 1:

Reviewer comments:

Reviewer #1

(Remarks to the Author)

This manuscript by Paouneskou et al addressed the important question of whether there are dedicated mechanisms that are necessary to “untether” chromosomes from the nuclear periphery during early meiosis, in the model organism *C. elegans*. In my original review, I was enthusiastic about the manuscript and raised a few points for the authors to clarify or provide additional support for some of their claims. I believe the authors have done an excellent job revising the manuscript to address my concerns. Additional data and thorough analysis have made the revised manuscript much improved. This work will be of great interest to the general audience studying the cell biology of the nucleus or genome organization, but will also be of particular interest to researchers in the field of meiosis, deepening our understanding of how chromosome dynamics is regulated during the dangerous but necessary cell cycle.

I appreciate the authors clarifying the timeline in their experiments. Yet there is one minor inconsistency remaining – presumably a typo: in the main text the authors stated “we depleted VRK-1 for 18h, and then transferred the worms to regular plates for approximately 48h. This strategy enabled VRK-1 to be expressed from pachynema onwards and throughout the embryonic divisions (Fig. 5a).” In the revised Fig. 5a, the timepoint after 18hr is 56hr, whereas $18 + 48 = 66$ hr.

Reviewer #2

(Remarks to the Author)

I congratulate the authors on their revised manuscript: they have performed several experiments and analyses to address the questions raised during the first evaluation which has strengthened the conclusions. I have only one concern regarding the conclusion on the effect of simultaneous depletion of VRK1 and BAF1 on chromosome attachment to the nuclear envelope (and point #4 in the first evaluation report). In the revised manuscript the authors state that “Depletion of both baf-1 and VRK-1 also released tethered chromosomes from the nuclear periphery in the -1 diakinesis oocyte (Fig. 2c, Supplementary Fig. 2d), whereas baf-1 depletion alone did not (Supplementary Fig. 2d).” However, as far as I can judge from the graph in Supplementary Fig. 2d, neither single BAF1 depletion nor double VRK1 BAF1 depletion has any effect compared to the control (“L4440 ethanol”).

Reviewer #3

(Remarks to the Author)

Reviewer #4

(Remarks to the Author)

The authors have addressed our concerns and the updated manuscript is a stronger, more compelling document.

We thank the reviewers for their enthusiasm for our study and the constructive suggestions to improve the manuscript.

REVIEWER COMMENTS

Reviewer #1 (Remarks to the Author):

In this work by Paouneskou et al., the authors addressed a long-standing question in meiosis – the mechanisms and roles of regulated chromosome tethering at the nuclear periphery during meiotic prophase – by combining induced protein degradation, chromosome painting, cytology, mutant/genetic analyses, and long-read sequencing. Meiosis depends on rapid chromosome movement in early prophase to achieve homolog pairing, synapsis, and eventually accurate segregation. In *C. elegans* – a premier experimental model system for studying meiosis – it's well established that the reorganization of chromosomes into a “crescent shaped” mass coincides with their rapid movement across the nuclear envelope (NE) in early prophase. However, the relationship between chromosome reorganization and their rapid movement remained unclear. Specifically, it remained enigmatic whether rapid chromosome movement was sufficient to “untether” chromosome from the NE, or additional mechanism might be required for this process. By focusing on a kinase (VRK-1) and the chromatin-NE tethering protein BAF-1, the authors identified a new mechanism that mediates the regulated tethering between chromosomes and the NE in early prophase. The authors first used auxin-inducible degradation to deplete VRK-1 and demonstrated that it caused “overtethering” of chromosomes in both leptotene/zygotene and diakinesis nuclei. They also demonstrated that the “overtethering” phenotype can be recapitulated by specific nonphosphorylatable mutations in BAF-1. They then carefully documented a series of meiotic phenotypes including delayed homolog pairing, synapsis, crossover formation, as well as elevated apoptosis upon VRK-1 depletion. Additionally, the authors showed that VRK-1 depletion led to aberrant chromosomes in maturing oocytes, in a manner that is dependent on BAF-1 and meiotic recombination. Finally, by taking advantage of the reversibility of the auxin inducible degradation system, the authors carried out an impressively meticulous assay to deplete VRK-1 only during transition zone and early pachynema and observed increased deletions and duplications in the next generation embryos. Overall this work was very well executed, with all the experiments well controlled. This work elucidated how VRK-1/BAF-1 regulates the tethering of chromosomes at the NE during meiosis, likely independent of rapid chromosome movement, with far-reaching implications for the roles of chromosome reorganization at the nuclear periphery in genome stability. Specific comments/ points are as follows. It should be noted that the points raised below did not dampen the high enthusiasm for this work and the perceived overall high interest and broad impact it will generate in the fields of meiosis, genetics, and cell biology of the nucleus.

(1) It would be helpful to clarify the localization of VRK-1, especially during leptotene/zygotene, where it was implicated to phosphorylate BAF-1 to “untether” chromosome from nuclear periphery. Do VRK-1 have more obvious NE localization in leptotene/zygotene nuclei compared to pachytene nuclei? From Supp Fig. 1c and Supp Fig. 5b, VRK-1 seem to have foci-like localization at the NE, however, this is obscured by the SUN-1(S8)Pi staining in the merged images in Supp Fig. 1c. It'd be informative to include monochromatic images and intensity measurement of VRK-1 in relation to an NE marker and/or chromosomes.

This point has been addressed in Supplementary Fig. 1c where we included monochromatic cytology images of VRK-1 (stained with HA), LMN-1 (NE marker) and DAPI (chromosomes). Additionally, we performed intensity measurements of VRK-1, LMN-1 and DAPI and we show that VRK-1 overlaps with DAPI and they are both enclosed by LMN-1 (NE- Supplementary Fig. 1d–e).

(2) The idea that chromosome movement is not affected by VRK-1 depletion (or BAF-1 phosphorylation) can be very interesting, because this could present a unique case where the “crescent shaped” chromosome organization is uncoupled from chromosome mobility during early meiosis. Have the authors looked at whether SUN-1 patches colocalize with chromosome pairing centers, especially upon VRK-1 depletion? This would be important to the conclusion that chromosome mobility can be uncoupled from VRK-1-mediated untethering from nuclear periphery. On a related note – have the author examined whether the length of SUN-1-Ser8Pi positive region change upon VRK-1 depletion?

We have addressed the colocalization SUN-1 patches and pairing centers by conducting costainings of SUN-1 S8Pi with PLK-2 antibodies with ethanol and auxin exposure (6h). In wild type, PLK-2 accumulates at the aggregates during the movement process and recruitment of PLK-2 is mediated by the pairing center proteins. This is shown in Figure 1d. The colocalization pattern between SUN-1 (S8Pi) and PLK-2 seems indistinguishable between the two conditions.

We also examined the length of the gonad region positive for SUN-1 (S8Pi) upon VRK-1 depletion and no significant difference was detected. This data is presented in Supplementary Fig. 1i.

(3) Per Supp Fig. 4f & 4g, it seems that the intrachromosome bridge phenotype can be caused by late/short depletion of VRK-1 alone (even from diplonema), whereas increased numbers of DAPI bodies require much longer depletion of VRK-1 (from transition zone). This suggests that intrachromosome bridge could be a separate phenotype specific to VRK-1/BAF-1's role during late prophase. Interestingly, VRK-1 depletion also caused

“overtethering” of chromosomes to the NE in diakinesis oocytes (Fig. 1j, where it says “18hr auxin” in the figure panel; yet it’s “50 hr” in the main text). Have the authors examined whether chromosome overtethering to the NE in “-1” oocytes can also happen with short auxin treatment (i.e. VRK-1 depletion from diplonema) that results in the bridge phenotype?

We have corrected the exposure time in the text to the actual 18h time point (this was a typo).

Regarding the bivalents at the periphery after 6 hours, we examined the “overtethering” of the chromosomes to the NE after 6 hours by performing the same analysis as in Fig. 1k–m. See Response Fig. 1 (below), even with only 6h of VRK-1 depletion, the bivalents are tethered to the periphery.

Response Fig. 1. Quantification of distance in diakinesis oocytes of the *vrk-1::degron::ha* between DAPI centroid and LMN-1 border after 6h in ethanol (n=128 centroids) or auxin (n=144 centroids). Mean ± SD ethanol vs auxin: 2.70 ± 1.00 vs 1.40 ± 0.57. P value < 0.0001 using Mann–Whitney test.

(4) The authors probably already have the data – how does *flag::baf-1-S4A* affect chromosome anchorage, bivalent number, or intrachromosome bridge in diakinesis oocytes?

Response Figures 2a and b below, shows show the quantification of the number of DAPI bodies and the number of bridges respectively in the two *Pi-baf-1* mutants compared to the *flag::baf-1* (wild type). *flag::baf-1^{S4A}* exhibits a significantly higher number of DAPI bodies as well as higher numbers of bridges in the -1 oocyte. This mutant has a severe and complex phenotype with long and short germlines—clearly defects from the progenitor zone get imported into meiosis. Additionally, the gonads are more fragile and as a result we did not proceed to a detailed quantification of the chromosome anchorage at the -1 oocyte, because we wanted to avoid a biased result which could lead to an overstatement.

Response Fig. 2 Quantification of number of DAPI bodies and bridges **a**, Mean ± S.D: *flag::baf-1* 6 ± 0 (n=37) vs *flag::baf-1^{T3A}* 5.97 ± 0.17 (n=33) (P=0.4076) and *flag::baf-1^{S4A}* 6.38 ± 0.67 P=0.0009 for both. **b**, Mean ± S.D: *flag::baf-1* 0 ± 0 (n=37) vs *flag::baf-1^{T3A}* 0.09 ± 0.29 (n=33) (P=0.0990) and *flag::baf-1^{S4A}* 0.66 ± 0.60 P<0.0001 for both

Minor points:

(1) It’d be helpful if the authors can clarify whether VRK-1 depletion in early meiosis, which caused increased deletions and duplications in the offspring, had any functional consequences. It was noted that after the 18 hr VRK-1 depletion, “embryo viability were similar to the wt following VRK-1 resynthesis (Supp Fig. 5b)”. Yet in Supp Fig. 5b, embryo viability seems to be indeed significantly lower than wt 2 days after VRK-1 depletion for 18hr, which appears to be the targeted time window according to Fig. 5a. If this is correct, the authors might consider highlighting it to underscore the physiological importance of VRK-1/BAF-1 regulated chromosome tethering. Also, have the authors checked male indices?

We have updated the viability graph in Supplementary Fig.5a with higher numbers of counts (n=10 per condition). The day 2, where the viability is indeed very significantly lower in the post-auxin recovered worms, is not the time-point we used to collect the embryos. We collected the embryos 63 hours later (they were allowed to be laid between 56-63 hours) which corresponds to the day 3, where the viability is much closer to the post-ethanol recovered worms. It is not surprising that the hatch rates are high since depleted oocytes are fertilized with wild-type sperm, therefore even if the rare genome variants would cause a mutation with a phenotype the animal would be heterozygous for that mutation.

We also checked for males when repeating the experiment and there was not a significant difference between the post-ethanol and post-auxin recovered worms (Supplementary Fig. 5b).

The viability counts are important to make sure that VRK-1 re-synthesis after depletion in transition zone is effective and early embryogenesis would occur in the presence of expressed VRK-1.

(2) In their introductory section discussing lamin's role in conferring "stability to germline nuclei", the authors may consider referencing to additional work such as [PMID: 37436986], [PMID: 21529718], and [PMID: 11071918], which could provide a fuller context and strengthen the scholarly background.

Thank you for this suggestion, we have included those references.

(3) The numbers in Supp Table 1 have a mixture of units (raw counts versus percentages), making comparisons (e.g. between different baf-1 mutants) confusing.

This has been corrected.

(4) In Acknowledgements and Table 4, Rueyling Lin's name was misspelled as Rueyling Kin.

This has been corrected.

Reviewer #2 (Remarks to the Author):

The manuscript by Paouneskou and coworkers reports that acute depletion of Vaccinia Related Kinase 1 (VRK1) causes multiple meiotic defects in the nematode *Caenorhabditis elegans*. VRK1 is essential and to control its expression, the protein is fused endogenously with a degron tag that is recognised by TIR1 in the presence of auxin, which triggers ubiquitination and degradation. By multiplexed FISH the authors demonstrate that chromosome organization is abnormal already within 6 hours of exposure to auxin and chromosomes fail to be released from the nuclear periphery of diakinesis oocytes after 18h. Barrier to Autointegration Factor 1 (BAF1) is a known VRK1 substrate and by mutation of two phosphorylation sites in BAF1, the authors demonstrate that the role of VRK1 in meiosis is at least partially through regulation of BAF1. Extended VRK1 depletion (by 50h auxin treatment) results in the formation of aberrant chromosomes, including fragmented chromosomes and intrachromosomal bridges. Finally, the authors show that a 18h pulse of VRK1 depletion followed by approximately 48h recovery leads to the formation of eggs with a higher frequency of chromosomal deletions and duplications. Altogether, the data represent a novel layer of regulation of chromosome behaviour during animal meiosis regulated by the protein kinase VRK1 and its substrate BAF1 at the nuclear envelope.

Technically, the study is very impressive and overall convincing. The quality of the microscopy images is impressive, and the manuscript is well-written. Nevertheless, the authors are recommended to consider the following points prior to publication:

1) The discrimination between clustered and non-clustered chromatin (e.g. Figures 1c and 2g) seems to be based on visual inspection. Because this is a subjective evaluation, the evaluator should not know the genotypes during scoring. Was this the case?

No, the quantified data were not blinded. However, we re-analyzed our cytology pictures by assessing the clustering using computational analysis which is more objective than visual inspection. We did this for Fig. 1b (Response Fig. 3a) and Fig. 2f (Response Fig. 3b).

As shown below, the computational analysis is in agreement with the visual assessment for both data sets. Some differences in significance in the left graph of Response Fig. 3b arose from a less efficient masking of the DAPI area in Cellpose which resulted in greater overlap between the DAPI and the SUN-1 (S8Pi) area.

Response Fig. 3. Computational analysis of clustering upon **a**, VRK-1 depletion, left graph: Mean \pm SD: 6h ethanol (n= 217 nuclei): 0.60 ± 0.01 vs 6h auxin (356 nuclei): 0.76 ± 0.13 , $P < 0.0001$ (Mann-Whitney), right graph: % Mean \pm SD: 6h ethanol (n= 9 gonads): $89.13 \% \pm 8.70$ vs 6h auxin (n= 10 gonads): $36.65 \% \pm 30.92$, $P = 0.0004$ (Mann-Whitney). **b**, upon BAF-1 phosphorylation on T3 or S4: left graph: Mean \pm SD: *flag::baf-1* (n=131 nuclei) 0.52 ± 0.12 vs *flag::baf-1^{T3A}* (n=150) 0.60 ± 0.12 , $P < 0.0001$ (Mann-Whitney) and *flag::baf-1^{S4A}* (n=201 nuclei) 0.72 ± 0.12 , $P < 0.0001$ compared to both (Mann-Whitney). Right graph: % Mean \pm SD: *flag::baf-1* (n=6 gonads) 84.88 ± 13.43 vs *flag::baf-1^{T3A}* (n=7 gonads) 62.84 ± 24.77 , $P = 0.1326$ and *flag::baf-1^{S4A}* (n=8 gonads) 24.75 ± 22.82 , $P = 0.0003$ vs *flag::baf-1* and 0.0318 vs *flag::baf-1^{T3A}* (P-values calculated with Kruskal-Wallis test).

Here is a detailed explanation of how this computational analysis was performed:

Regions of interest (ROIs), corresponding to the first ten rows of germ cells after meiotic entry, were manually defined using Adobe Photoshop. Individual fluorescence channels (DAPI and SUN1) were split and saved as .tif files for downstream analysis.

Cell segmentation was performed using Cellpose version 4, applying optimized but consistent parameters across all samples. Each channel (DAPI, SUN1) was processed separately using channel-specific parameters. The segmentation masks were saved as .tif images.

In Python, segmented objects were analyzed using the scikit-image library. For each SUN1-labeled object, the area was measured, and the overlap with DAPI was computed to determine nuclear inclusion. Only SUN1 regions with >30% DAPI overlap and above a minimum size threshold were retained. For accepted objects, the ratio of DAPI-overlapping area to total SUN1 area was calculated.

Results were compiled into a .csv file and imported into GraphPad Prism for statistical analysis and visualization.

Response Fig.4. Methodology and workflow for the computational analysis of datasets in Fig. 1c and 2g.

2) Page 5. Based on Figure 1e, the authors state that depletion of VRK1 causes increased association of chromosomes to the nuclear periphery. However, this should be quantified (or wait with the claim until describing the data in Figure 1j-l).

This has been addressed by rephrasing to: "High resolution imaging of the three chromosomes in the transition zone suggested a more pronounced association to the nuclear periphery after auxin treatment".

3) According to Figure 1l, depletion of VRK1 causes chromosome bivalents to accumulate at the nuclear periphery in diakinesis nuclei. According to the main text, the experiment was performed by exposure to auxin for 50 h but the figure indicates 18h. Please clarify. Moreover, have shorter time points been tested? Fig S1c indicate that VRK-1 is efficiently depleted from the pachynema zone after only 6 h: from this zone the chromosomes condense shortly after.

The timepoint in the text has been corrected to 18h (this was a typo).

We also tested a shorter timepoint of 6h, which resulted in the same significant chromosome anchorage to the periphery upon VRK-1 depletion. This has also been brought up by reviewer 1, below the quantification.

Response Fig. 1 (because it has been already shown earlier). Quantification of distance in diakinesis oocytes of the *vrk-1::degron::ha* between DAPI centroid and LMN-1 border after 6h in ethanol (n=128 centroids) or auxin (n=144 centroids). Mean \pm SD ethanol vs auxin: 2.70 ± 1.00 vs 1.40 ± 0.57 . P value < 0.0001 using Mann-Whitney test.

4) On page 7 the authors claim that "Depletion of both baf-1 and VRK-1 also released tethered chromosomes from the nuclear periphery in the -1 diakinesis oocyte (Fig. 2c), whereas baf-1 depletion alone did not." This should be quantified (only a single image is shown).

We did the quantification and this is now shown in Supplementary Figure 2d. To mark the border of the nucleus, we used SUN-1 (S8Pi) and the analysis was done exactly like the Fig. 1k-m. The method used for quantification has been updated in the methods part as well.

5) The data in Figure 2d-e indicate an unexpected increase in nuclear BAF1 signal upon VRK1 depletion. How can this increase be explained? BAF1 is usually a nuclear protein so changes in nucleocytoplasmic transport is unlikely to be the explanation. Change in transcription and/or protein stability? Is this also observed more acutely, for instance after 6 h of exposure to auxin?

We repeated this experiment for 6h time point and the staining we acquired was of higher quality. Thus, we replaced the previous cytology insets with the new set (Figure 2d) and we have also included the full germline immunostained for FLAG in ethanol (+VRK-1) and auxin conditions (-VRK-1) (Supplementary Fig. 2e). We performed intensity quantifications which showed a slight increase of BAF-1 in the mitotic zone and a very strong increase in transition zone upon VRK-1 depletion (Fig. 2e). BAF-1 accumulation appears strongest at the nuclear rim, however nuclear BAF-1 is also increased upon VRK-1 depletion. Phosphorylation of BAF-1 might indeed increase the turnover of the protein-perhaps when it is not phosphorylated it is stuck to the chromosomes.

6) On page 7 the authors argue that "Chromatin reorganization was impaired in both [baf1] mutants". However, the T3A mutant seems to behave as the wild type in all assays. Related to this, did the authors consider to simultaneously mutate both residues?

We have rephrased this sentence to "The percentage of nuclei with clustered chromatin in the first 10 cell rows after meiotic entry was similar for both *flag::baf-1* and *flag::baf-1^{T3A}*, but significantly lower in *flag::baf-1^{S4A}*". We did not manage to recover a mutant with both T3A and S4A, even with several attempts. We hypothesize that this is because of strong developmental defects caused by the S4A mutation alone and the double mutant does not give rise to an oocyte. To obtain the S4A mutation, we had to inject approx. 150 worms from this we isolated only one line carrying the S4A mutation.

7) The interpretation of the Western blot depicted in Figure 2h is not obvious. Firstly, there seems to be less

material in the nuclear fractions in the auxin sample compared to the ethanol sample (e.g. less H3 and FLAG signals), which may contribute to the already weak purple band indicated with the left white arrow to become invisible (right white arrow). Secondly, multiple purple bands seem not to be recognised by the FLAG antibody, which is unexpected. Finally, from the Methods section, it is not clear if this analysis was performed multiple times, i.e. if it is reproducible. Information on how the antibody was generated is also missing. For instance, will it recognise BAF phosphorylated both on T3 and S4 or only when phosphorylated exclusively on S4?

The extra bands that are not recognized by the FLAG antibody are unspecific bands coming from the BAF-1 (S4Pi) antibody (it is possible that a similar motif in another protein is recognized?). We have now included a Western blot using whole worm extracts from the following strains: *flag::baf-1*, *flag::baf-1^{T3A}*, *flag::baf-1^{S4A}* (Supplementary Fig. 2i). This time we used actin as a loading control. The band detected by the BAF-1 (S4Pi) antibody in the *flag::baf-1* and *flag::baf-1^{T3A}* was absent in the *flag::baf-1^{S4A}*, indicating the specificity of the antibody (full blots are included in the submission). It also indicates that the antibody is specific for the S4 phospho-modification. In the material and methods, we also added information about the generation of the antibody by Eurogentec.

8) Regarding the Nanopore sequencing: The authors state that “insertions and inversions seem to be evenly distributed along the genome”. In worms, attachment to the nuclear periphery is mainly through the distal ends of chromosomes. If the role of VRK1 is through phosphorylation of BAF1 and release of chromosome (distal ends), perhaps one could expect to see an increase in rearrangements in these regions? Or do the authors propose that the location of lesions is independent of distance to the anchoring to the nuclear periphery through BAF1?

We hypothesize that the chromosome motions affect the chromosomes along their entire length and that lack of movement does not discourage ectopic contacts along the entire chromosome.

9) Can the authors (roughly) estimate how many deletions and duplications were observed per cell or per embryo? The VRK1-depleted embryos have more mutations but normal viability according to the text, which seems surprising.

Yes, It is possible to get a rough estimate of the number of duplications and deletions per embryo. However, as we expect a large number of false positives from the SV callers (especially as we had to relax the calling parameters to detect low frequency variants), the results are likely an overestimation. Below are the assumptions, the steps we followed, and the estimates for both deletions and duplications:

We had 40,000 ethanol-treated eggs and 30,000 auxin-treated eggs, and the sequencing coverage after matching the read length distributions for both libraries was around 600. The sperm was made before auxin-treatment, so all the embryos with SVs are expected to be heterozygous. Assuming all the detected SVs are real:

Total DNA molecules per sample = #embryos x #cells/embryo x ploidy
Total DNA molecules ethanol-treated = 40,000 x 10 x 2 = 800,000 DNA molecules
Total DNA molecules auxin-treated = 30,000 x 10 x 2 = 600,000 DNA molecules
Sequencing depth (DP) = 600

SV allele Frequency = read support/Sequencing depth

For the SVs that have 1 to 2 read support, the frequency in the original sample should be around:

$1/DP \leq SV \text{ Frequency} \leq 2/DP$

The number of occurrences of an SV supported by 1 to 2 reads in the original sample for it to be detected should then be between:

$1/DP \times \text{Total DNA molecules per sample} \leq \text{SV occurrences} \leq 2/DP \times \text{Total DNA molecules per sample}$

For ethanol-treated:

$1/600 \times 800,000 \leq \text{SV occurrences} \leq 2/600 \times 800,000$

1333.3 DNA molecules \leq SV occurrences \leq 2666.7 DNA molecules

For auxin-treated:

$1/600 \times 600,000 \leq \text{SV occurrences} \leq 2/600 \times 600,000$

1000 DNA molecules \leq SV occurrences \leq 2000 DNA molecules

We can now estimate the #embryos carrying SV in the original sample (assuming they are heterozygous for the wildtype and that they have 10 cells):

#embryos carrying variant = (SV occurrences X ploidy) / (#cells/embryo)

For ethanol-treated:

$(1333 \times 2) / 10 \leq \text{\#embryos carrying variant} \leq (2666.7 \times 2) / 10$

266 embryos \leq #embryos carrying variant \leq 533 embryos

For auxin-treated:

$(1000 \times 2) / 10 \leq \text{\#embryos carrying variant} \leq (2000 \times 2) / 10$

200 embryos \leq #embryos carrying variant \leq 400 embryos

To estimate the number of variants per embryo, we can then:

#variants per embryo = (#SVs x #embryos carrying variant) / #embryos

For instance, the number of deletions in the ethanol control is 2456 and in auxin treated 5633:
 $(2456 \times 266)/40,000 \leq \# \text{variants per embryo in ethanol} \leq (2456 \times 533)/40,000$
 16 deletions $\leq \# \text{deletions per embryo in ethanol} \leq 32.7262$ deletions

$(5633 \times 200)/40,000 \leq \# \text{variants per embryo in auxin} \leq (5633 \times 400)/40,000$
 28 deletions $\leq \# \text{variants per embryo in ethanol} \leq 56.33$ deletions

Review table 1: estimated number of deletions and duplications per embryo for the ethanol and auxin treated embryos.

	Total variants (ethanol)	#variants per embryo ethanol	Total variants (auxin)	#variants Per embryo auxin
Deletions	2456	Between 16 and 32.7	5633	Between 28.165 and 56.3
Duplications	688	Between 4.58 and 9.17	2080	Between 10.4 and 20.8

Regarding the viability, the fairly normal viability in VRK1-depleted embryos at the day of collection (Day 3 in the Supp. Figure 5a) can possibly be explained by all the embryos being heterozygous due the fertilization by wildtype sperm. The other possibility is that a large number of SVs fall in either intergenic or intronic regions and are not significantly enriched in gene-rich regions, which is what the variant affect analysis using VEP suggests (Response Fig. 5-6).

Response Fig. 5a-c: Variant affect analysis for the duplications of the different sizes revealed that the duplications detected fall vastly in intergenic or intronic regions

Response Fig. 6a-c: Variant affect analysis for the deletions of the different sizes revealed that the duplications detected fall vastly in intergenic or intronic regions

10) General point: many graphs report multiple statistical analyses within the same graph and experiment, but it is not clear if p values were adjusted accordingly. Some p values are borderline to be considered significant and may be affected by such adjustment.

We thank the reviewer for noticing this. In the previously called Figures 1h and j we had included multiple statistical analyses. After the reviewer's comment, we again performed the statistical analysis applying the Holm-Šídák method for the adjustments of the p-values. For the data set of previous Figure 1h and the number of Y shapes, the adjusted P- value is 0.1142, thus the difference is not significant anymore. However, we would like to point out that this is because of the size effect of our dataset. Applying the Cliff's test still showed a small effect in for the number of Y shapes upon VRK-1 depletion (Cliff's delta=0.131). A data set for the statistical analysis would need to be larger than what we can do with this very tedious assay. Due to the difficulties with this assay and time constrains to increase our N number, we decided to remove this data since the same scientific content is also transported with the single locus Fish analysis ("**pairing is delayed**"). We therefore moved the 5S Fish analysis from the supplements into Figure 1. For the previously called Fig. 1j (now called 1h), we also applied the Holm- Šídák method which calculated an adjusted P-value of 0.0101 which maintained the difference between the overlapping of LGII-LGIII highly significant. We have included also the adjusted P- values for LGI-LGII and LGI-LGIII in the main text and figure legend. For the newly added Supplementary Fig. 2d, we performed the Kruskal-Wallis Test for multiple comparisons and for the correction (adjustment of the P-values) we applied the

Dunn's method.

Minor points:

Page 4. The sentence “*vrk-1(syb2608 vrk-1::AID::ha); tir-1::mRuby* (hereafter called *vrk-1::AID::ha*)” is incomplete and needs revision. The authors should mention in which tissue and/or when during development TIR1 is expressed.

This has been re-phrased to: Therefore, we employed an auxin-inducible degradation (AID) system to investigate the effects of VRK-1 depletion at the different meiotic stages of prophase [25], and generated the *vrk-1(syb2608 vrk-1::AID::ha); tir-1::mRuby* (hereafter called *vrk-1::AID::ha*) strain. In this strain, *tir-1* is under the control of the *sun-1* promoter, and with this is expressed in the germline and early embryos.

It would be relevant to show examples of microscope images of EdU pulse labelling to complement the cartoons. We do not really find a good spot to add this data. Here it is:

Response Fig. 7: Panel of *vrk-1::AID::ha* germlines stained for EdU and DAPI at time 0h (upper germline), 18h (middle germline) and 48h later (lower germline). Scale bar: 10 μ m.

Chromosomes I, II and III were analysed by multiplexed FISH. Why were these chromosomes analysed and not IV, V and X?

The protocol we are using is based on the paper of Fields *et al.*, 2019 which labels with single colour probes (red, green and far red) chromosome I, II and II; whereas IV, V and X are labelled with combinations of colors (IV: red+green, V: red+far red, X: green and far red). The painting of the 6 chromosome is working in our hands. However, for data analysis, we had to restrict ourselves to focus on 3 colour labelling to measure the overlap of these chromosomes in 3 dimensions. It was too confusing to tell the chromosomes apart otherwise.

What does the arrowhead in Figure 3b indicate?

It highlights the unsynapsed region (HTP-3 signal without SYP-1). We added this to the figure legend.

Please revise the percentages in Figure 4e: $1/30 = 3.3\%$ (not 0.03%); $18/30 = 60\%$ (not 62.5%).

This has been corrected.

Figure 4f-g: Why was this done for 48h (vs 50h for the rest of Fig4) and from L4 (vs from day 1 adult for the rest of Fig 4)?

The reason for this is the employment of RNAi. RNAi was more efficient when worms were incubated as L4s and for 48 hours.

Regarding the apoptosis analysis, the authors should comment that this observation is opposite to the situation in a *vrk-1* loss of function allele (Waters et al, 2010).

Waters et al, examined a different developmental context. They showed that VRK-1 counteracted a cell cycle arrest mediated through *cep-1/p53*. In their depletion assays the gonad might not contain cells that are even competent to undergo apoptosis because of the strong proliferation and/or developmental defect. We depleted VRK-1 in non-proliferating meiocytes and here no influence of VRK-1 was observed on *cep-1* mediated apoptosis.

Supplementary Table 1: The left column reports (partial) genotypes but not strain names as the title indicates.

This has been corrected.

Reviewer #3 (Remarks to the Author):

Reviewer #4 (Remarks to the Author):

The authors of this mostly well-executed and rigorous study exploits the auxin induced degradation of VRK-1, an essential protein, during meiotic prophase. Using acute depletion of VRK-1, the authors show that a characterized VRK-1/BAF-1 axis that ensures that chromosomes are removed from the nuclear periphery with entry into mitosis plays a similar essential role during meiosis, actively regulating the scheduled removal of chromosomes from the nuclear periphery during entry into meiotic prophase. Thus, this work demonstrates that this removal is an active process that is not merely accomplished by the chromosome mobility that is a conserved aspect of meiotic prophase, a novel finding. They show that this dispersal does not have consequences on chromosome movement but does have consequences on pairing, synapsis and recombination. Moreover they show that this axis takes advantage of a conserved phosphorylation site on BAF-1, serine 4. The most impactful part of the paper, aside from the identification of this mechanism, is the authors' demonstration that the inability to remove chromosomes from the nuclear periphery upon entry into meiotic prophase has dramatic consequences for genomic stability, where deletions, duplications, insertions and inversions are increased when this removal is prevented by acute depletion of VRK-1. This result shows that this regulated dispersal likely limits ectopic recombination and ensures that meiotic recombination and crossover formation occurs correctly, maintaining genome stability during sexual reproduction. I do not have the expertise to comment on the sequencing approach. However, I have a few major concerns about the other experiments, mostly controls, that need to be addressed.

Major points

1. Throughout the paper, the authors use the term chromatin organization. Given the role that tethering may play in regulating gene expression in interphase, I understand this choice. But using the term chromatin organization raises questions about histone modifications, transcriptional activity, etc. I suggest the authors use the term chromosome organization instead to describe the changes they see upon VRK-1 depletion.

It has been changed to chromosome organisation.

2. In general, the authors could make their last point, about the increase in structural chromosomes rearrangements when VRK-1 is depleted in meiosis, stronger. That they can infer this from their cytological analysis and show it with their sequencing analysis is powerful. This work makes a compelling argument that this mechanism helps reorganize chromosomes in the nucleus upon entry into meiotic prophase and has profound consequences on genome stability, especially since chromosome mobility at PC ends seems normal. Given previous data that defects in bouquet formation increases ectopic recombination in yeast (Goldman and Lichten, 2000), this result supports a model where chromosomes have to be removed from the nuclear periphery to facilitate pairing, synapsis and accurate meiotic recombination and maintain genomic stability during a dangerous but necessary event, meiotic recombination.

We thank this reviewer for acknowledging the importance of our work. We included a sentence at the end of the introduction.

3. The authors could also explain details about the *C. elegans* germline better to make the study more accessible to people in other fields. For example, the authors should point out the spatio-temporal organization of meiotic nuclei, especially with respect to their analysis of synapsis and recombination in Figure 3. Similarly, the authors could explain better that the connected bivalents are the products of recombination in their analysis in Figure 4.

This has been addressed in the text.

4. The section on biochemical experiments needs to be expanded, to explain why and how this fractionation technique was undertaken and additional experiments are necessary, including important controls. Moreover, I could not find a description of the methods used to generate the phospho-specific antibody.

This fractionation experiment has been performed twice. The detection of the BAF-1 (S4Pi) has been performed once. All the blots probed for the HA, GAPDH, FLAG and BAF-1 (S4Pi) are included as part of the submission (Source data). Detailed protocols for obtaining the fractions and the workflow is included in the material and methods (Nuclei isolation and protein fractionation from large auxin-treated and control *C. elegans* cultures).

What happens when they use the anti-phospho Ser4 antibody on germlines or lysates from *baf-1S4A* mutants? This is a necessary control to ensure the specificity of the antibody, especially given the signal that is still visible in Figure 2h in auxin-treated worms (see below).

We have added the relevant western blot in Supplementary Fig. 2i where we used whole worm extracts from the following strains: *flag::baf-1*, the *flag::baf-1^{T3A}* and the *flag::baf-1^{S4A}*. The BAF-1 (Ser4Pi) band was missing only in the *flag::baf-1^{S4A}* extract supporting the specificity of the antibody.

Although I agree that there I can't detect phosphorylated Baf-1 in the nuclear insoluble fraction in in auxin-treated VRK-1::AID::HA worms, why is there still phosphorylated Baf-1 in the cytosolic and nuclear soluble fractions? Is there another kinase that phosphorylates Baf-1 in these fractions?

A blot is provided in the Supplementary Fig. 2i (with whole worm extracts of the *baf-1* phospho-mutants to show that the antibody does recognize BAF-1 S4Pi). VRK-1 depletion is driven by the expression of the *sun-1* promoter and it is possible that depletion is not 100% effective or some cells were included where *sun-1* is not efficiently expressed. It is also possible that different fractions of BAF-1 in different cellular compartments are accessible to different kinases.

- Shouldn't there be more Baf-1 associated with chromatin in auxin-treated VRK-1::AID::HA strains, especially given their immunofluorescence data in Figure 2d and e? If the authors think this discrepancy is because only specific regions of the germline exhibit this enrichment upon treatment with auxin, they should explicitly say that.

We have up-dated this figure.

- Can the authors provide the methods for generating anti-phospho Ser4 antibody?

It has been added in the Materials and Methods. The antibody was generated by Eurogentec following the standardized procedure of the company.

5. Are the authors surprised to see the number of structural variants in their ethanol controls? Does this say something about the functionality of the VRK-1::AID::HA transgene, even if it supports viability? Or does it say something about the mutagenic potential of meiotic recombination? Would you see similar numbers of structural variants in wildtype animals? This experiment is not strictly required but some commentary would be useful.

Although we agree that there is a possibility that the auxin is leaky (there is almost no strain where this is not seen), it is more likely that a large number of the structural variants observed in both the ethanol control and auxin treated embryos are false positives, which is why we focused on the fold difference (at least 2 fold change difference when comparing ethanol and auxin), as it is more likely to reflect biological differences. The SV callers are known to have precision <95% ^{1,2}, and as we had to allow for the detection of low frequency variants (SVs that are supported by 1 or 2 reads), we expect the precision to be much lower. We have tested simulating nanopore reads with the same coverage, median read length, 0% heterozygosity, and 99% accuracy, with no structural variants from the *C. elegans* genome using Sim-it (<https://github.com/ndierckx/Sim-it?tab=readme-ov-file>) and performing the same analysis. The analysis recovered many structural variants >1000 bp with 1 to 2 read support, suggesting the callers are prone to generating false positives.

Response table 2: the number of SVs called by cuteSV and sniffles using simulated reads as input.

	cuteSV	sniffles
DEL	448	369
DUP	253	425
INV	253	268
INS	8	45
BND	697	170

Minor:

1. The authors should spell out wildtype instead of using "wt."

It has been addressed and rephrased to wild type.

2. Can the authors use a different color scheme for some Figures that make images easier to see, such as Figure 2D? The yellow and magenta on white is particularly difficult to see.

It is made like that for colour blind purposes. We have provided the monochromatic blots of the FLAG and the BAF-1 (S4Pi) antibodies.

3. The authors try to tie their results back to what has been seen in other systems as a way to showcase conservation, but those examples could be written to be more clear and more explicit in their connections.

We tried to focus on presenting our results; broader discussions with other literature (broader than we already did) is not really feasible since we cannot increase the number of references (we are already at the recommended limit).

4. Can the authors indicate the Pairing Center end of chromosomes when reporting on the frequency of structural variants (Figures 5F and G and Supplementary Figures 9-20)?

We have indicated in Fig. 5f, g and Supplementary Fig. 9-20, the region where the Pairing Center regions are located. We used Phillips et al. 2009³ for guidance.

5. The authors should check their spelling. For example, Needhi Bhalla is spelled incorrectly (Nheedi Bhalla)

It has been corrected.

References

- 1 Helal, A. A., Saad, B. T., Saad, M. T., Mosaad, G. S. & Aboshanab, K. M. Benchmarking long-read aligners and SV callers for structural variation detection in Oxford nanopore sequencing data. *Sci Rep* **14**, 6160 (2024), <https://doi.org/10.1038/s41598-024-56604-2>
- 2 Jiang, T., Liu, Y., Jiang, Y., Li, J., Gao, Y., Cui, Z., Liu, Y., Liu, B. & Wang, Y. Long-read-based human genomic structural variation detection with cuteSV. *Genome Biol* **21**, 189 (2020), <https://doi.org/10.1186/s13059-020-02107-y>
- 3 Phillips, C. M., Meng, X., Zhang, L., Chretien, J. H., Urnov, F. D. & Dernburg, A. F. Identification of chromosome sequence motifs that mediate meiotic pairing and synapsis in *C. elegans*. *Nat Cell Biol* **11**, 934-942 (2009), <https://doi.org/10.1038/ncb1904>

We thank the reviewers for their careful reading of our manuscript and for pointing out the last two ambiguous points.

REVIEWERS' COMMENTS

Reviewer #1 (Remarks to the Author):

I appreciate the authors clarifying the timeline in their experiments. Yet there is one minor inconsistency remaining – presumably a typo: in the main text the authors stated "we depleted VRK-1 for 18h, and then transferred the worms to regular plates for approximately 48h. This strategy enabled VRK-1 to be expressed from pachynema onwards and throughout the embryonic divisions (Fig. 5a)." In the revised Fig. 5a, the timepoint after 18hr is 56hr, whereas $18 + 48 = 66$ hr.

We have corrected the inconsistency in the text. We corrected the 48h to 45 and also clarified the time interval of embryo collection in more detail. We also corrected the timeline in Fig. 5a.

Reviewer #2 (Remarks to the Author):

In the revised manuscript the authors state that "Depletion of both baf-1 and VRK-1 also released tethered chromosomes from the nuclear periphery in the -1 diakinesis oocyte (Fig. 2c, Supplementary Fig. 2d), whereas baf-1 depletion alone did not (Supplementary Fig. 2d)." However, as far as I can judge from the graph in Supplementary Fig. 2d, neither single BAF1 depletion nor double VRK1 BAF1 depletion has any effect compared to the control ("L4440 ethanol").

Single depletion of BAF-1 did not affect chromosomal anchorage to the periphery and that's why it is not different from the control. The double depletion of BAF-1 and VRK-1 released the chromosomal anchorage to the periphery that would be induced by VRK-1 depletion. That is the reason why there is not significant difference when compared to the wild type. We have made this explanation and rephrased the text.